# CycleNet: Enhancing Time Series Forecasting through Modeling Periodic Patterns

**Shengsheng Lin**[1], **Weiwei Lin**[1,2,*], **Xinyi Hu**[3], **Wentai Wu**[4], **Ruichao Mo**[1], **Haocheng Zhong**[1]

[1]School of Computer Science and Engineering, South China University of Technology, China
[2]Pengcheng Laboratory, China
[3]Department of Computer Science and Engineering, The Chinese University of Hong Kong
[4]College of Information Science and Technology, Jinan University, China
`cslinshengsheng@mail.scut.edu.cn`, `linww@scut.edu.cn`, `xyhu@cse.cuhk.edu.hk`,
`wentaiwu@jnu.edu.cn`, `{cs_moruichao, cshczhong}@mail.scut.edu.cn`

## Abstract

The stable periodic patterns present in time series data serve as the foundation for conducting long-horizon forecasts. In this paper, we pioneer the exploration of explicitly modeling this periodicity to enhance the performance of models in long-term time series forecasting (LTSF) tasks. Specifically, we introduce the Residual Cycle Forecasting (RCF) technique, which utilizes learnable recurrent cycles to model the inherent periodic patterns within sequences, and then performs predictions on the residual components of the modeled cycles. Combining RCF with a Linear layer or a shallow MLP forms the simple yet powerful method proposed in this paper, called CycleNet. CycleNet achieves state-of-the-art prediction accuracy in multiple domains including electricity, weather, and energy, while offering significant efficiency advantages by reducing over 90% of the required parameter quantity. Furthermore, as a novel plug-and-play technique, the RCF can also significantly improve the prediction accuracy of existing models, including PatchTST and iTransformer. The source code is available at: https://github.com/ACAT-SCUT/CycleNet.

## 1 Introduction

Time series forecasting (TSF) plays a crucial role in various domains such as weather forecasting, transportation, and energy management, providing insights for early warnings and facilitating proactive planning. Particularly, accurate predictions over long horizons (e.g., spanning several days or months) offer increased convenience, referred to as Long-term Time Series Forecasting (LTSF) [59, 56, 17, 42, 6]. However, the principle enabling long-horizon prediction lies in understanding the inherent periodicity within the data [32]. Unlike short-term forecasting, long-term predictions cannot rely solely on recent temporal information (including means, trends, etc.). For instance, a user's electricity consumption thirty days ahead not only correlates with their consumption patterns in the past few days.

In such cases, long-term dependencies, or in other words, underlying stable periodicity within the data, serve as the practical foundation for conducting long-term predictions [32]. This is why existing models emphasize their capability to extract features with long-term dependencies. Models like Informer [59], Autoformer [51], and PatchTST [40] utilize the Transformer's ability for long-distance modeling to address LTSF tasks. ModernTCN [38] employs large convolutional kernels to enhance TCNs' ability to capture long-range dependencies, and SegRNN [31] uses segment-wise iterations to

---

*Corresponding author

38th Conference on Neural Information Processing Systems (NeurIPS 2024).

improve RNN methods' handling of long sequences. If a model can accurately capture long-range dependencies, it can precisely extract periodic patterns from historical long sequences, enabling more accurate long-horizon predictions.

However, if the purpose of constructing deep and complex models is solely to better extract periodic features from long-range dependencies, why not directly model the patterns? As illustrated in Figure 1, electricity data exhibits clear daily periodic patterns (in addition to possible weekly patterns). We can use a globally shared daily segment to represent the periodic pattern in electricity consumption. By repeating this daily segment $N$ times, we can continuously represent the cyclic components of $N$ days' electricity consumption sequences.

Based on the above motivation, we pioneer explicit modeling of periodic patterns in the data to enhance the model's performance on LTSF tasks in this paper. Specifically, we propose the **Residual Cycle Forecasting** (RCF) technique. It involves using *learnable recurrent cycles* to explicitly model the inherent periodic patterns within time series data, followed by predicting the residual components of the modeled cycles. Combining the RCF technique with either a single-layer Linear or a dual-layer MLP results in **CycleNet**, a simple yet powerful method. CycleNet achieves consistent state-of-the-art performance across multiple domains and offers significant efficiency advantages.

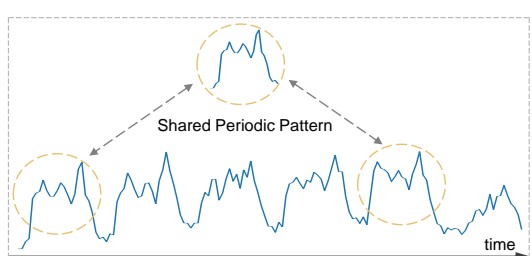

Figure 1: Shared daily periodic patterns present in the Electricity dataset.

In summary, this paper contributes:

- We identify the presence of shared periodic patterns in long-horizon forecasting domains and propose explicit modeling of these patterns to enhance the model's performance on LTSF tasks.

- Technically, we introduce the RCF technique, which utilizes learnable recurrent cycles to explicitly model the inherent periodic patterns within time series data, followed by predicting the residual components of the modeled cycles. The RCF technique significantly enhances the performance of basic (or existing) models.

- Applying RCF with a Linear layer or a shallow MLP forms the proposed simple yet powerful method, called CycleNet. CycleNet achieves consistent state-of-the-art performance across multiple domains and offers significant efficiency advantages.

## 2 Related work

In fact, utilizing periodic information to enhance model prediction accuracy is not a novel concept. Numerous studies, in particular, have introduced a series of Seasonal-Trend Decomposition (STD) techniques that allow models to better leverage periodic information. Popular models such as Autoformer [51], FEDformer [60], and DLinear [56] utilize the classical STD approach to decompose the original time series into two equally sized subsequences: seasonal and trend components, which are then modeled independently. These classical STD methods typically use a basic moving average (MOV) kernel to perform a sliding aggregation to obtain the trend component. Recently, Leddam [55] proposed replacing the traditional MOV kernel in STD with a Learnable Decomposition (LD) kernel, leading to improved performance. Additionally, DEPTS [8] treats the periodicity of sequences as a parameterized function with respect to time, and learns periodic and residual components layer-wise through its periodic and local blocks. SparseTSF [32], another recent work, utilizes cross-period sparse forecasting technique to decouple cycles and trends, achieving impressive performance at extremely low cost.

The RCF technique proposed in this paper can essentially be considered a type of STD method. The key difference from existing techniques lies in its explicit modeling of global periodic patterns within independent sequences using learnable recurrent cycles. The proposed RCF technique is *conceptually simple*, *computationally efficient*, and yields *significant improvements* in prediction accuracy. The

further proposed CycleNet, which combines the RCF technique with a simple backbone, is a Linear-
or MLP-based model that is simple, efficient, and powerful for time series forecasting. To correctly
position CycleNet, we have provided a detailed review of the development of different categories of
time series forecasting methods (including Transformer-based, RNN-based, etc.) in Appendix A.

## 3 CycleNet

Given a time series $X$ with $D$ variables or channels, the objective of time series forecasting is to
predict future horizons $H$ steps ahead based on past $L$ observations, mathematically represented as
$f : x_{t-L+1:t} \in \mathbb{R}^{L \times D} \to \bar{x}_{t+1:t+H} \in \mathbb{R}^{H \times D}$. In fact, the inherent periodicity within time series
is fundamental for accurate prediction, particularly when forecasting over large horizons, such as
96-720 steps (corresponding to several days or months). To enhance the model's performance on
long-term prediction tasks, we propose the Residual Cycle Forecasting (RCF) technique. It combines
a Linear layer or a shallow MLP to form a simple yet powerful method CycleNet, as illustrated in
Figure 2, with detailed pseudocode provided in Appendix B.1.

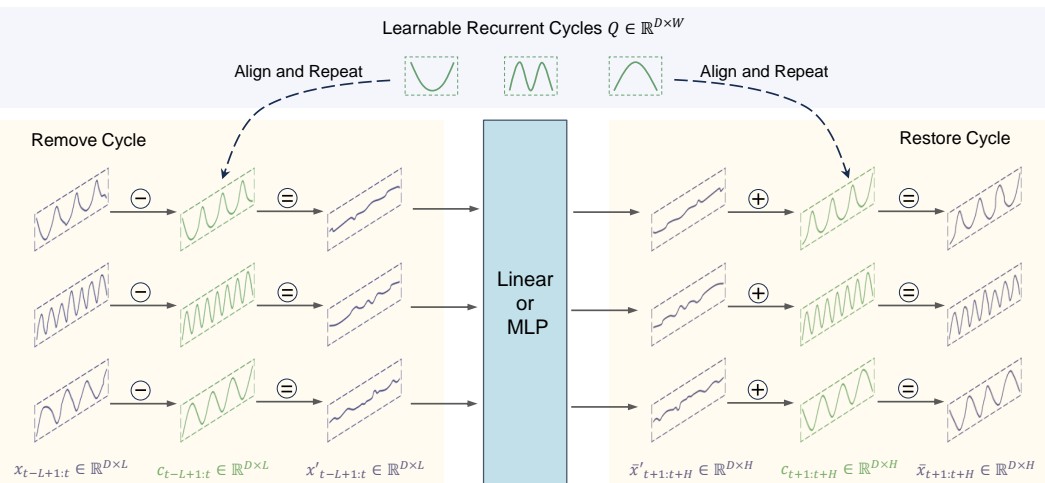

Figure 2: CycleNet architecture. CycleNet/Linear and CycleNet/MLP represent using a single-layer
Linear model and a dual-layer MLP model, respectively, as the backbone of CycleNet. Here, $D = 3$.

### 3.1 Residual cycle forecasting

The RCF technique comprises two steps: the first step involves modeling the periodic patterns of
sequences through learnable recurrent cycles within independent channels, and the second step entails
predicting the residual components of the modeled cycles.

**Periodic patterns modeling**   Given $D$ channels with a priori cycle length $W$, we first generate
*learnable recurrent cycles* $Q \in \mathbb{R}^{W \times D}$, all initialized to zeros. These recurrent cycles are globally
shared within channels, meaning that by performing cyclic replications, we can obtain cyclic com-
ponents $C$ of the sequence $X$ of the same length. These recurrent cycles $Q$ of length $W$ undergo
gradient backpropagation training along with the backbone module for prediction, yielding learned
representations (distinct from the originally initialized zeros) that unveil the internal cyclic patterns
within the sequence.

Here, the cycle length $W$ depends on the a priori characteristics of the dataset and should be set to the
maximum stable cycle within the dataset. Considering that scenes requiring long-term predictions
usually exhibit prominent, explicit cycles (e.g., electrical consumption and traffic data exhibit clear
daily and weekly cycles), determining the specific cycle length is available and straightforward. Ad-
ditionally, the dataset's cycles can be further examined through autocorrelation functions (ACF) [39],
as revealed in Appendix B.2.

**Residual forecasting**   Predictions made on the residual components of the modeled cycles, termed residual forecasting, are as follows:

1. Remove the cyclic components $c_{t-L+1:t}$ from the original input $x_{t-L+1:t}$ to obtain residual components $x'_{t-L+1:t}$.

2. Pass $x'_{t-L+1:t}$ through the backbone to obtain predictions for the residual components, $\bar{x}'_{t+1:t+H}$.

3. Add the predicted residual components $\bar{x}'_{t+1:t+H}$ to the cyclic components $c_{t+1:t+H}$ to obtain $\bar{x}_{t+1:t+H}$.

It is important to note that, since the cyclic components $C$ are virtual sequences derived from the cyclic replications of $Q$, we cannot directly obtain the aforementioned sub-sequences $c_{t-L+1:t}$ and $c_{t+1:t+H}$. Therefore, as illustrated in Figure 3, appropriate alignments and repetitions of the recurrent cycles $Q$ are needed to obtain equivalent sub-sequences: (i) Left-shift $Q$ by $t \bmod W$ positions to obtain $Q^{(t)}$. Here, $t \bmod W$ can be viewed as the relative positional index of the current sequence sample within $Q$. (ii) Repeat $Q^{(t)}$ $\lfloor L/W \rfloor$ times and concatenate $Q^{(t)}_{0:L \bmod W}$. Mathematically, these two equivalent subsequences can be represented as:

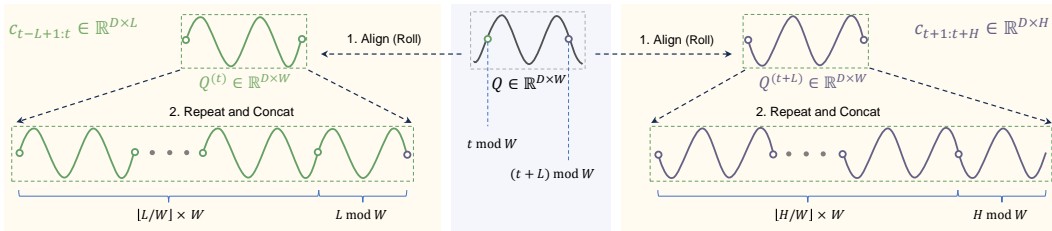

Figure 3: Alignments and repetitions of the recurrent cycles $Q$. Here, $D = 1$.

$$c_{t-L+1:t} = [\underbrace{Q^{(t)}, \cdots, Q^{(t)}}_{\lfloor L/W \rfloor}, Q^{(t)}_{0:L \bmod W}], \tag{1}$$

$$c_{t+1:t+H} = [\underbrace{Q^{(t+L)}, \cdots, Q^{(t+L)}}_{\lfloor H/W \rfloor}, Q^{(t+L)}_{0:H \bmod W}]. \tag{2}$$

**Backbone**   The original prediction task is transformed into cyclic residual component modeling, which can serve as normal sequence modeling. Therefore, any existing time series forecast model can be employed as a backbone. In this paper, our aim is to propose and examine a method for enhancing time series prediction by explicitly modeling cycles (i.e., RCF). Thus, we opt for the most basic backbone, namely a single-layer Linear and a dual-layer MLP, forming our simple yet powerful methods, CycleNet/Linear and CycleNet/MLP. Herein, each channel utilizes the same backbone with parameter sharing for modeling, which is also referred to as the Channel Independent strategy [13].

### 3.2   Instance normalization

The statistical properties of time series data, such as the mean, often vary over time, which is referred to as distributional shifts. This can lead to poor performance of models trained on historical training sets when applied to future data. To address this issue, recent research has introduced Instance Normalization strategies like RevIN [45, 22, 26]. Mainstream approaches such as iTransformer [37], PatchTST [40], and SparseTSF [32] have widely adopted similar techniques to enhance performance. To improve the robustness of CycleNet, we also incorporate a similar *optional* strategy (see the full ablation study in Appendix C.4). Specifically, we remove the varying statistical properties from the model's internal representations outside of CycleNet's input and output steps:

$$x_{t-L+1:t} = \frac{x_{t-L+1:t} - \mu}{\sqrt{\sigma + \epsilon}}, \tag{3}$$

$$\bar{x}_{t+1:t+H} = \bar{x}_{t+1:t+H} \times \sqrt{\sigma + \epsilon} + \mu, \tag{4}$$

where $\mu$ and $\sigma$ represent the mean and standard deviation of the input window, respectively, and $\epsilon$ is a small constant for numerical stability. This method aligns with the RevIN version that excludes learnable affine parameters [22].

### 3.3 Loss function

To remain consistent with current mainstream methods, CycleNet defaults to using Mean Squared Error (MSE) as the loss function to ensure fair comparison with other methods, formulated as:

$$\mathcal{L}oss = \|x_{t+1:t+H} - \bar{x}_{t+1:t+H}\|_2^2. \tag{5}$$

## 4 Experiments

### 4.1 Setup

**Datasets**  We utilized widely adopted benchmark datasets including the ETT series [59], Weather, Traffic, Electricity, and Solar-Energy [24]. Preprocessing operations on the datasets, such as dataset splitting and normalization methods, remained consistent with prior works (e.g., Autoformer [51], iTransformer [37], etc.).

The information of the datasets is shown in Table 1. Note that these datasets all exhibit stable cyclic patterns, such as daily and weekly, which form the realistic basis for performing long-horizon forecasting. Combined with the sampling frequency of the datasets, we can infer the maximum cycle length of the datasets, such as 24 for ETTh1 and 168 for Electricity. These manually inferred cycle lengths can be further confirmed through the ACF analysis, details of which are provided in Appendix B.2. The hyperparameter $W$ of CycleNet is set by default to match the cycle length in Table 1.

Table 1: Dataset Information.

| Dataset | ETTh1 & ETTh2 | ETTm1 & ETTm2 | Electricity | Solar-Energy | Traffic | Weather |
|---|---|---|---|---|---|---|
| Timesteps | 17,420 | 69,680 | 26,304 | 52,560 | 17,544 | 52,696 |
| Channels | 7 | 7 | 321 | 137 | 862 | 21 |
| Frequency | 1 hour | 15 mins | 1 hour | 10 mins | 1 hour | 10 mins |
| Cyclic Patterns | Daily | Daily | Daily & Weekly | Daily | Daily & Weekly | Daily |
| **Cycle Length** | **24** | **96** | **168** | **144** | **168** | **144** |

**Baselines**  We compared CycleNet against state-of-the-art models in recent years, including iTransformer [37], PatchTST [40], Crossformer [58], TiDE [5], TimesNet [52], DLinear [56], SCINet [34], FEDformer [60], Autoformer [51]. To comprehensively evaluate CycleNet's performance, the Mean Squared Error (MSE) and Mean Absolute Error (MAE) metrics were employed.

**Environments**  All experiments in this paper were implemented using PyTorch [41], trained using the Adam [23] optimizer, and executed on a single NVIDIA GeForce RTX 4090 GPU with 24 GB memory.

### 4.2 Main results

Table 2 shows the comparison results of CycleNet with other models on multivariate LTSF tasks. Overall, CycleNet achieves state-of-the-art performance (except for the Traffic dataset), with CycleNet/MLP ranking first overall, and CycleNet/Linear ranking second overall. Due to the nonlinear

mapping capability of MLP compared to Linear, CycleNet/MLP performs better on high-dimensional datasets such as Electricity and Solar-Energy (i.e., datasets with more than 100 channels). In summary, with the support of the RCF technique, even a very simple and basic model (i.e., Linear and MLP) can achieve the current best performance, surpassing other deep models. This fully demonstrates the advantages of the RCF technique.

Table 2: Multivariate long-term time series forecasting results. The look-back length $L$ is fixed as 96 and the results are averaged from all prediction horizons of $H \in \{96, 192, 336, 720\}$. Full results and more comparison results on *longer* look-back lengths are available in Appendix C.2. The results of other models are sourced from iTransformer [37] and TimeMixer [48]. The best results are highlighted in **bold** and the second best are underlined.

| Dataset | ETTh1 | | ETTh2 | | ETTm1 | | ETTm2 | | Electricity | | Solar-Energy | | Traffic | | Weather | |
|---|---|---|---|---|---|---|---|---|---|---|---|---|---|---|---|---|
| Metric | MSE | MAE | MSE | MAE | MSE | MAE | MSE | MAE | MSE | MAE | MSE | MAE | MSE | MAE | MSE | MAE |
| Autoformer [2021] | 0.496 | 0.487 | 0.450 | 0.459 | 0.588 | 0.517 | 0.327 | 0.371 | 0.227 | 0.338 | 0.885 | 0.711 | 0.628 | 0.379 | 0.338 | 0.382 |
| FEDformer [2022] | 0.440 | 0.460 | 0.437 | 0.449 | 0.448 | 0.452 | 0.305 | 0.349 | 0.214 | 0.327 | 0.291 | 0.381 | 0.610 | 0.376 | 0.309 | 0.360 |
| SCINet [2022] | 0.747 | 0.647 | 0.954 | 0.723 | 0.485 | 0.481 | 0.571 | 0.537 | 0.268 | 0.365 | 0.282 | 0.375 | 0.804 | 0.509 | 0.292 | 0.363 |
| DLinear [2023] | 0.456 | 0.452 | 0.559 | 0.515 | 0.403 | 0.407 | 0.350 | 0.401 | 0.212 | 0.300 | 0.330 | 0.401 | 0.625 | 0.383 | 0.265 | 0.317 |
| TimesNet [2023] | 0.458 | 0.450 | 0.414 | 0.427 | 0.400 | 0.406 | 0.291 | 0.333 | 0.192 | 0.295 | 0.301 | 0.319 | 0.620 | 0.336 | 0.259 | 0.287 |
| TiDE [2023] | 0.541 | 0.507 | 0.611 | 0.550 | 0.419 | 0.419 | 0.358 | 0.404 | 0.251 | 0.344 | 0.347 | 0.417 | 0.760 | 0.473 | 0.271 | 0.320 |
| Crossformer [2023] | 0.529 | 0.522 | 0.942 | 0.684 | 0.513 | 0.496 | 0.757 | 0.610 | 0.244 | 0.334 | 0.641 | 0.639 | 0.550 | 0.304 | 0.259 | 0.315 |
| PatchTST [2023] | 0.469 | 0.454 | 0.387 | 0.407 | 0.387 | 0.400 | 0.281 | 0.326 | 0.205 | 0.290 | 0.270 | 0.307 | 0.481 | 0.304 | 0.259 | 0.281 |
| TimeMixer [2024] | 0.447 | 0.440 | **0.364** | **0.395** | 0.381 | **0.395** | 0.275 | 0.323 | 0.182 | 0.272 | **0.216** | 0.280 | 0.484 | 0.297 | **0.240** | **0.271** |
| iTransformer [2024] | 0.454 | 0.447 | 0.383 | 0.407 | 0.407 | 0.410 | 0.288 | 0.332 | 0.178 | 0.270 | 0.233 | 0.262 | **0.428** | **0.282** | 0.258 | 0.278 |
| CycleNet/Linear | **0.432** | **0.427** | 0.383 | 0.404 | 0.386 | **0.395** | 0.272 | 0.315 | 0.170 | 0.260 | 0.235 | 0.270 | 0.485 | 0.313 | 0.254 | 0.279 |
| CycleNet/MLP | 0.457 | 0.441 | 0.388 | 0.409 | **0.379** | 0.396 | **0.266** | **0.314** | **0.168** | **0.259** | **0.210** | **0.261** | 0.472 | 0.301 | 0.243 | **0.271** |

Furthermore, we can observe that CycleNet's performance on the Traffic dataset is inferior to iTransformer, which models multivariate relationships in time series data using an inverted Transformer. This is because the Traffic dataset exhibits spatiotemporal characteristics and temporal lag characteristics, where the traffic flow at a certain detection point significantly affects the future values of neighboring detection points. In such cases, modeling sufficient inter-channel relationships is necessary, and iTransformer accomplishes this. In contrast, CycleNet independently models the temporal dependencies of each channel, hence it suffers a disadvantage in this scenario. However, CycleNet still significantly outperforms other baselines on the Traffic dataset, demonstrating the competitiveness of CycleNet. Additionally, we have included more analysis of CycleNet in traffic scenarios in Appendix C.5, including a full comparison of results on the PEMS datasets.

### 4.3 Efficiency analysis

The proposed RCF technique, as a plug-and-play module, requires minimal overhead, needing only additional $W \times D$ learnable parameters and no additional Multiply-Accumulate Operations (MACs). The backbones of CycleNet, namely single-layer Linear and dual-layer MLP, are also significantly lightweight compared to other multi-layer stacked models. Table 3 demonstrates the efficiency comparison between CycleNet and other mainstream models, where CycleNet shows significant advantages. Particularly, compared to iTransformer, which also possesses strong capabilities in modeling long-term dependencies and nonlinear learning, CycleNet/MLP has over ten times fewer parameters and MACs. As for CycleNet/Linear, which shares the same single-layer linear backbone as

Table 3: Efficiency comparison between CycleNet and other models on the Electricity dataset with look-back length $L = 96$ and forecast horizon $H = 720$. Training Time denotes the average time required per epoch for the model.

| Model | Parameters | MACs | Training Time(s) |
|---|---|---|---|
| Informer [2021] | 12.53M | 3.97G | 70.1 |
| Autoformer [2021] | 12.22M | 4.41G | 107.7 |
| FEDformer [2022] | 17.98M | 4.41G | 238.7 |
| DLinear [2023] | 139.6K | 44.91M | 18.1 |
| PatchTST [2023] | 10.74M | 25.87G | 129.5 |
| iTransformer [2024] | 5.15M | 1.65G | 35.1 |
| CycleNet/MLP | 472.9K | 134.84M | 30.8 |
| CycleNet/Linear | 123.7K | 22.42M | 29.6 |
| RCF part | 53.9K | 0 | 12.8 |

DLinear, it also has fewer parameters and MACs. However, in terms of training speed, DLinear is still faster than CycleNet/Linear. This is because the RCF technique requires aligning the recurrent cycles with each data sample, which incurs additional CPU time. Overall, considering the significant improvement in prediction accuracy brought by the RCF technique, CycleNet achieves the best balance between performance and efficiency.

## 4.4 Ablation study and analysis

**Effectiveness of RCF**    To investigate the effectiveness of RCF, we conducted comprehensive ablation experiments on two datasets with significant periodicity: Electricity and Traffic. The results are shown in Table 4.

Table 4: Ablation study of RCF technique. The Linear and MLP backbones apply the same instance normalization strategy as CycleNet by default to fully demonstrate the effect of RCF technique.

| Dataset | Electricity | | | | | | | | Traffic | | | | | | | |
|---|---|---|---|---|---|---|---|---|---|---|---|---|---|---|---|---|
| Horizon | 96 | | 192 | | 336 | | 720 | | 96 | | 192 | | 336 | | 720 | |
| Metric | MSE | MAE | MSE | MAE | MSE | MAE | MSE | MAE | MSE | MAE | MSE | MAE | MSE | MAE | MSE | MAE |
| Linear | 0.197 | 0.274 | 0.197 | 0.277 | 0.212 | 0.292 | 0.253 | 0.324 | 0.645 | 0.383 | 0.598 | 0.361 | 0.605 | 0.362 | 0.643 | 0.381 |
| + RCF | 0.141 | 0.234 | 0.155 | 0.247 | 0.172 | 0.264 | 0.210 | 0.296 | 0.480 | 0.314 | 0.482 | 0.313 | 0.476 | 0.303 | 0.503 | 0.320 |
| Improve | **28.6%** | **14.6%** | **21.4%** | **10.8%** | **18.8%** | **9.5%** | **17.1%** | **8.7%** | **25.6%** | **18.0%** | **19.5%** | **13.2%** | **21.3%** | **16.2%** | **21.8%** | **16.1%** |
| MLP | 0.175 | 0.259 | 0.181 | 0.265 | 0.197 | 0.282 | 0.240 | 0.317 | 0.500 | 0.325 | 0.496 | 0.321 | 0.509 | 0.325 | 0.542 | 0.342 |
| + RCF | 0.136 | 0.229 | 0.152 | 0.244 | 0.170 | 0.264 | 0.212 | 0.299 | 0.458 | 0.296 | 0.457 | 0.294 | 0.470 | 0.299 | 0.502 | 0.314 |
| Improve | **22.2%** | **11.6%** | **15.9%** | **8.0%** | **13.6%** | **6.3%** | **11.6%** | **5.7%** | **8.5%** | **8.9%** | **7.9%** | **8.3%** | **7.7%** | **8.0%** | **7.3%** | **8.1%** |
| DLinear | 0.195 | 0.278 | 0.194 | 0.281 | 0.207 | 0.297 | 0.243 | 0.331 | 0.649 | 0.398 | 0.599 | 0.372 | 0.606 | 0.375 | 0.646 | 0.396 |
| + RCF | 0.143 | 0.240 | 0.156 | 0.253 | 0.171 | 0.270 | 0.204 | 0.302 | 0.506 | 0.317 | 0.499 | 0.317 | 0.512 | 0.325 | 0.545 | 0.343 |
| Improve | **26.6%** | **13.6%** | **19.7%** | **10.0%** | **17.4%** | **8.9%** | **16.3%** | **8.8%** | **22.1%** | **20.4%** | **16.6%** | **14.6%** | **15.4%** | **13.3%** | **15.6%** | **13.5%** |
| PatchTST | 0.168 | 0.260 | 0.176 | 0.266 | 0.193 | 0.282 | 0.233 | 0.317 | 0.436 | 0.281 | 0.449 | 0.285 | 0.464 | 0.293 | 0.499 | 0.310 |
| + RCF | 0.136 | 0.231 | 0.153 | 0.246 | 0.170 | 0.264 | 0.211 | 0.299 | 0.438 | 0.264 | 0.457 | 0.270 | 0.469 | 0.275 | 0.509 | 0.292 |
| Improve | **19.0%** | **11.0%** | **13.0%** | **7.6%** | **11.7%** | **6.6%** | **9.4%** | **5.7%** | -0.5% | **6.1%** | -1.8% | **5.5%** | -1.0% | **6.3%** | -2.0% | **6.1%** |
| iTransformer | 0.148 | 0.240 | 0.162 | 0.253 | 0.178 | 0.269 | 0.225 | 0.317 | 0.395 | 0.268 | 0.417 | 0.276 | 0.433 | 0.283 | 0.467 | 0.302 |
| + RCF | 0.136 | 0.231 | 0.153 | 0.247 | 0.168 | 0.263 | 0.194 | 0.287 | 0.415 | 0.263 | 0.440 | 0.271 | 0.456 | 0.278 | 0.491 | 0.294 |
| Improve | **8.1%** | **3.7%** | **5.6%** | **2.4%** | **5.8%** | **2.2%** | **13.8%** | **9.5%** | -5.1% | **1.9%** | -5.5% | **1.8%** | -5.3% | **1.8%** | -5.1% | **2.6%** |

Firstly, when combining the basic Linear and MLP backbones (both utilizing instance normalization by default) with the RCF technique, a significant improvement in prediction accuracy (approximately 10% to 20%) is observed. This demonstrates that the success of CycleNet is largely attributed to the RCF technique rather than the backbones themselves or the instance normalization strategy. Overall, the performance of MLP is stronger than that of Linear, regardless of whether the RCF technique is applied. This indicates that non-linear mapping capability is necessary when modeling high-dimensional datasets with the channel-independent strategy (sharing parameters across each channel), aligning with previous research findings [26].

Secondly, we further verified whether RCF can enhance the prediction accuracy of existing models, as RCF is essentially a plug-and-play flexible technique. It is observed that incorporating RCF still improves the performance of existing complex designed, deep stacked models (approximately 5% to 10%), such as PatchTST [40] and iTransformer [37]. Even for DLinear, which already employs the classical MOV-based STD technique, RCF was able to provide an improvement of approximately 20%. This further indicates the effectiveness and portability of RCF.

However, an interesting phenomenon was observed: although the MAE decreases when PatchTST and iTransformer are combined with RCF, the MSE increases. The most important reason behind this is that there are extreme points in the Traffic dataset that could affect the effectiveness of RCF, which fundamentally relies on learning the historical average cycles in the dataset. We further analyze this phenomenon in detail in Appendix C.5 and suggest potential directions for improving the RCF technique.

**Comparison of different STD techniques**    The proposed RCF technique is essentially a more powerful STD approach. Unlike existing methods that decompose the periodic (seasonal) component from a limited look-back window, RCF learns the global periodic component from the training set. Here, we compare the effectiveness of RCF with existing STD techniques, using a pure Linear model as the backbone (without applying any instance normalization strategies). The comparison includes LD from Leddam [55], MOV from DLinear [56], and Sparse technique from SparseTSF [32]. As shown in Table 5, RCF significantly outperforms other STD methods, particularly on datasets with strong periodicity, such as Electricity and Solar-Energy. In contrast, the other STD methods did not show significant advantages over the pure Linear model.

There are several reasons for this. First, MOV and LD-based STD methods achieve trend estimation by sliding aggregation within the look-back window, which suffers from inherent issues [27, 26]: (i) The sliding window of the moving average needs to be larger than the maximum period of the seasonal component; otherwise, the decomposition may be incomplete (especially when the period length exceeds the look-back sequence length, making decomposition potentially impossible).

(ii) Zero-padding is required at the edges of the sequence samples to obtain equally sized moving average sequences, leading to distortion of the sequence edges. As for the Sparse technique, being a lightweight decomposition method, it relys more on longer look-back windows and instance normalization strategies to ensure adequate performance.

Additionally, these methods that decouple trend and seasonality within the look-back window are essentially equivalent to unconstrained or weakly constrained linear regression [44], which means that after full training convergence, linear-based models combined with these methods are theoretically equivalent to pure linear models. In contrast, the periodic components obtained by the RCF technique are globally estimated from the training set, allowing it to surpass the limitations of a finite-length look-back window, and thus, its capabilities extend beyond standard linear regression.

Table 5: Comparison of different STD techniques. To directly compare the effects of STD, the configuration used here is consistent with that of DLinear [56], with a sufficient look-back window length of 336 and no additional instance normalization strategies. Thus, CLinear here refers to CycleNet/Linear without RevIN. The reported results are averaged across all prediction horizons of $H \in \{96, 192, 336, 720\}$, with full results available in Appendix C.3.

| Setup | CLinear (RCF+Linear) | | LDLinear (LD+Linear) | | DLinear (MOV+Linear) | | SLinear (Sparse+Linear) | | Linear | |
|---|---|---|---|---|---|---|---|---|---|---|
| Metric | MSE | MAE | MSE | MAE | MSE | MAE | MSE | MAE | MSE | MAE |
| ETTh1 | **0.418** | **0.434** | 0.427 | 0.439 | 0.425 | 0.437 | 0.424 | 0.436 | 0.427 | 0.439 |
| ETTh2 | **0.451** | **0.456** | 0.455 | 0.457 | 0.471 | 0.467 | 0.460 | 0.460 | 0.460 | 0.462 |
| ETTm1 | **0.349** | **0.382** | 0.365 | 0.387 | 0.367 | 0.390 | 0.362 | 0.383 | 0.362 | 0.384 |
| ETTm2 | **0.266** | **0.330** | 0.273 | 0.336 | 0.280 | 0.341 | 0.290 | 0.352 | 0.269 | 0.331 |
| Electricity | **0.157** | **0.255** | 0.167 | 0.264 | 0.167 | 0.264 | 0.172 | 0.268 | 0.167 | 0.265 |
| Solar-Energy | **0.220** | **0.259** | 0.253 | 0.316 | 0.254 | 0.318 | 0.255 | 0.315 | 0.253 | 0.318 |
| Traffic | **0.423** | **0.289** | 0.434 | 0.296 | 0.434 | 0.296 | 0.435 | 0.292 | 0.434 | 0.296 |
| Weather | 0.245 | 0.300 | 0.244 | 0.297 | **0.244** | **0.296** | 0.246 | 0.298 | 0.245 | 0.297 |

**Impact of hyperparameter $W$** The hyperparameter $W$ determines the length of the learnable recurrent cycles $Q$ in the RCF technique. In principle, it must match the maximum primary cycle length in the data to correctly model the periodic patterns of the sequence. We investigate the performance of the CycleNet/Linear model under different settings of $W$ for different datasets in

Table 6: Performance of the CycleNet/Linear model with varied $W$. The forecast horizon is set as 96.

| Setup | RCF/W=168 | | RCF/W=144 | | RCF/W=96 | | RCF/W=24 | | W/o. RCF | |
|---|---|---|---|---|---|---|---|---|---|---|
| Metric | MSE | MAE | MSE | MAE | MSE | MAE | MSE | MAE | MSE | MAE |
| Electricity | **0.142** | **0.234** | 0.196 | 0.275 | 0.196 | 0.274 | 0.195 | 0.274 | 0.197 | 0.274 |
| Traffic | **0.480** | **0.314** | 0.617 | 0.386 | 0.617 | 0.385 | 0.618 | 0.385 | 0.645 | 0.383 |
| Solar-Energy | 0.289 | 0.376 | **0.208** | **0.256** | 0.276 | 0.365 | 0.287 | 0.375 | 0.286 | 0.375 |
| ETTm1 | 0.350 | 0.369 | 0.340 | 0.366 | **0.325** | **0.363** | 0.348 | 0.367 | 0.351 | 0.372 |
| ETTh1 | 0.395 | 0.402 | 0.384 | 0.395 | 0.383 | 0.393 | **0.377** | **0.391** | 0.384 | 0.392 |

Table 6. When correctly setting the hyperparameter $W$ to the max cycle length of the dataset (i.e., the cycle length pre-inferred in Table 1), RCF can play a significant role, yielding a large performance gap compared to the cases when it is not correctly set. This indicates the necessity of inferring and setting the correct $W$ for RCF to function properly. Furthermore, when $W$ is incorrectly set, the model's performance is almost the same as when RCF is not used at all. This suggests that even in the worst-case scenario, RCF does not bring significant negative effects.

**Visualization of the learned periodic patterns** The purpose of the RCF technique is to utilize the learnable recurrent cycles $Q$ (initialized to zero) to model the periodic patterns in time series data. After co-training with the backbone, the recurrent cycles can represent the inherent periodic patterns of the sequence. Figure 4 illustrates the different periodic patterns learned from different datasets and channels. For example, Figure 4(c) shows the daily operating pattern of solar photovoltaic generation, while Figure 4(d) displays the weekly operating pattern of traffic flow, featuring peak traffic in the mornings on weekdays. These periodic patterns learned from the global sequence provide important supplementary information to the prediction model, especially when the length of the look-back window is limited and may not provide sufficient cyclic information when the cycle length is long.

Furthermore, although the cycle length is the same for different channels within the same dataset, the specific periodic patterns differ, as shown in Figure 4(e-h). Particularly, Figure 4(f) demonstrates the intermittent periodicity of household electricity consumption on weekdays, while others exhibit relatively uniform weekday patterns in their respective channels. This highlights the necessity of separately modeling the periodic patterns for each channel.

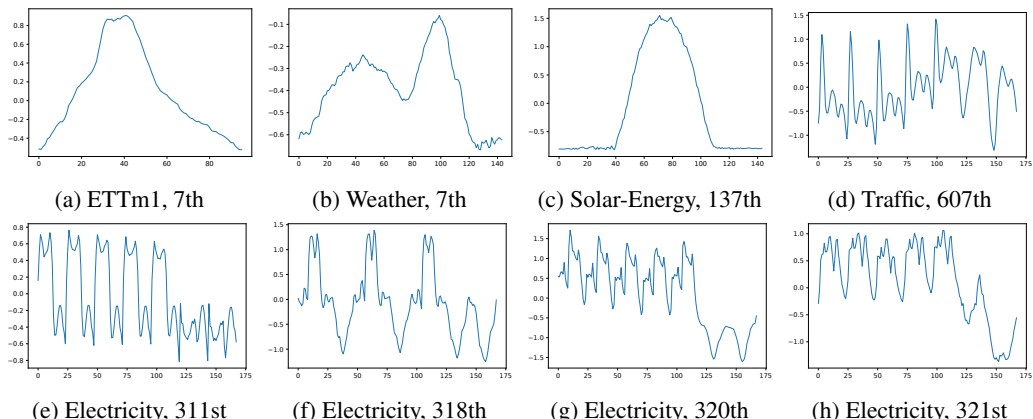

(a) ETTm1, 7th      (b) Weather, 7th      (c) Solar-Energy, 137th      (d) Traffic, 607th

(e) Electricity, 311st      (f) Electricity, 318th      (g) Electricity, 320th      (h) Electricity, 321st

Figure 4: Visualization of the periodic patterns learned by CycleNet/Linear. Panels (a-d) display different periodic patterns learned from different datasets, and panels (e-h) show different periodic patterns learned from different channels within the same dataset. The $i$ th indicates the index of the channel within the dataset.

In conclusion, these findings demonstrate that the RCF technique can effectively learn the inherent periodic patterns in time series data, serving as a crucial explanatory factor contributing to the state-of-the-art performance of CycleNet. Additionally, we have included further analysis in Appendix C.1, showcasing the learned periodic patterns of RCF under different configurations to better illustrate how RCF operates.

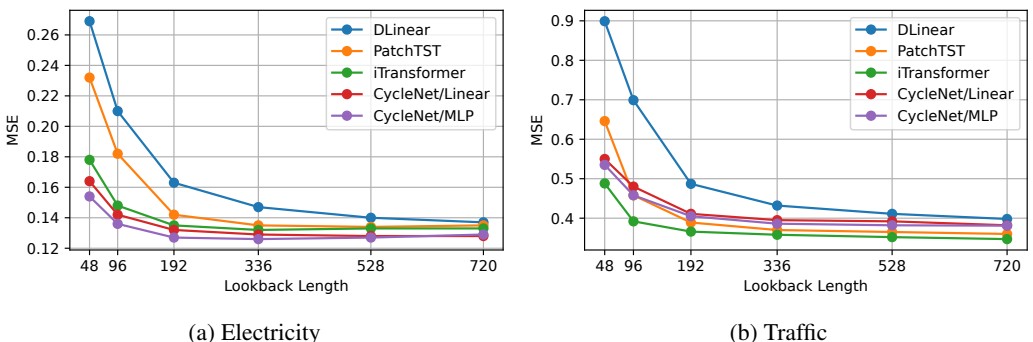

(a) Electricity                           (b) Traffic

Figure 5: Performance of CycleNet and comparative models with different look-back lengths. The forecast horizon is set as 96.

**Performance with varied look-back length** The look-back length determines the richness of historical information that can be utilized. Theoretically, the larger it is, the better the model performance should be, especially for models capable of capturing long-term dependencies. Figure 5 shows the performance of different models under different look-back lengths. It can be observed that CycleNet, as well as representatives of current state-of-the-art models such as iTransformer [37], PatchTST [40], and DLinear [56], all achieve better performance with longer look-back lengths. This indicates that these models all possess strong capabilities in modeling long-term dependencies.

It is worth highlighting that (i) on the Electricity dataset, CycleNet outperforms current state-of-the-art models at any prediction length; (ii) on the Traffic dataset, CycleNet still falls short compared to powerful existing multivariate forecasting models, such as iTransformer. This indicates that in scenarios with strong periodicity but without additional spatiotemporal relationships, fully leveraging the periodic components is sufficient to achieve high-accuracy predictions. However, in more complex scenarios that require thorough modeling of relationships between variables, a simple channel-independent strategy combined with a basic backbone, like CycleNet, still struggles to fully meet the demands. Therefore, in Appendix C.5, we further analyze the current limitations of the current RCF technique in spatiotemporal scenarios (such as the traffic domain) and and point out

potential directions for future improvements. Finally, we also provide a comparison of CycleNet with existing models on full datasets using longer look-back windows in Appendix C.2.

## 5 Discussion

**Potential limitations**  CycleNet demonstrates its efficacy in LTSF scenarios characterized by prominent and explicit periodic patterns. However, there are several potential limitations of CycleNet that warrant discussion here:

- **Unstable cycle length:** CycleNet may not be suitable for datasets where the cycle length (or frequency) varies over time, such as electrocardiogram (ECG) data, because CycleNet can only learn a fixed-length cycle.

- **Varying cycle lengths across channels:** When different channels within a dataset exhibit cycles of varying lengths, CycleNet may encounter challenges because it defaults to modeling all channels with the same cycle length $W$. Given CycleNet's channel-independent modeling strategy, one potential solution is to pre-process the dataset by splitting it based on cycle lengths or to independently model each channel as a separate dataset.

- **Impact of outliers:** If the dataset contains significant outliers, CycleNet's performance may be affected. This is because the fundamental working principle of RCF is to learn the historical average cycles in the dataset. When significant outliers exist, the mean of a certain point in the cycle learned by RCF can be exaggerated, leading to inaccurate estimation of both the periodic and residual components, which subsequently impacts the prediction process.

- **Long-range cycle modeling:** The RCF technique is effective for modeling mid-range stable cycles (e.g., daily or weekly). However, considering longer dependencies (such as yearly cycles) presents a more challenging task for the RCF technique. Although, in theory, CycleNet's $W$ can be set to a yearly cycle length to model annual cycles, the biggest difficulty lies in collecting sufficiently long historical data to train a complete yearly cycle, which might require decades of data. In this case, future research needs to develop more advanced techniques to specifically address long-range cycle modeling.

**Future work: further modeling inter-channel relationships**  The RCF technique enhances the model's ability to model the periodicity of time series data but does not explicitly consider the relationships between multiple variables. In some spatio-temporal scenarios where spatial and temporal dependencies between variables exist, these relationships are crucial. For example, recent studies such as iTransformer [37] and SOFTS [12] indicate that appropriately modeling inter-channel relationships can improve performance in traffic scenarios. However, directly applying the RCF technique to iTransformer does not lead to significant improvement (at least for the MSE metric), as demonstrated in Table 4. We believe that devising a more reasonable multivariate modeling approach that combines CycleNet could be promising and valuable, and we leave it for future exploration.

## 6 Conclusion

This paper reveals the presence of inherent periodic patterns in time series data and pioneers the exploration of explicitly modeling this periodicity to enhance the performance of time series forecasting models. Technically, we propose the Residual Cycle Forecasting (RCF) technique, which models the shared periodic patterns in sequences through recurrent cycles and predicts the residual cyclic components via a backbone. Furthermore, we introduce the simple yet powerful LTSF methods CycleNet/Linear and CycleNet/MLP, which combine single-layer Linear and dual-layer MLP respectively with the RCF technique. Extensive experiments demonstrate the effectiveness of the RCF technique, and CycleNet as a novel and simple method achieves state-of-the-art results with significant efficiency advantages. The findings in this paper underscore the importance of periodicity as a key characteristic for accurate time series prediction, which should be given greater emphasis in the modeling process. Finally, integrating CycleNet with effective inter-channel relationship modeling methods serves as a promising and valuable future research direction.

## Acknowledgments

This work is supported by Guangdong Major Project of Basic and Applied Basic Research (2019B030302002), National Natural Science Foundation of China (62072187), Guangzhou Development Zone Science and Technology Project (2023GH02) and the Major Key Project of PCL, China under Grant PCL2023A09.

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

# A  Development of time series forecasting

In recent years, the time series analysis community has shifted its focus from short-term forecasting to tasks with longer prediction horizons, also known as LTSF tasks . This shift offers greater convenience but also poses increased challenges. Mainstream approaches can be roughly classified into the following five distinct classes:

**Transformer-based Models**    It is widely recognized that Transformers possess impressive capabilities for long-distance modeling, and thus researchers have high expectations for their adaptation to long time series tasks [46, 49]. Early research works, such as LogTrans [25], TFT [29], Informer [59], Autoformer [51], Pyraformer [35], FEDformer [60], ETSformer [50], and NSTransformer [36], focused on optimizing the original Transformer architecture for time series analysis tasks. However, more recent research has found that satisfactory performance can be achieved by simply partitioning patches, drawing inspiration from patch techniques used in the computer vision community [7, 14]). Approaches like PatchTST [40], PETformer [30], and Crossformer [58] have demonstrated promising results by adopting this patch-based approach.

**Linear- and MLP-based Models**    Linear- and MLP-based methods are often lighter-weight, especially compared to Transformer methods that require stacking multiple blocks [57, 4].  A particularly notable breakthrough is the observation made by DLinear [56], which demonstrates that a single-layer linear approach could outperform many complex Transformer designs. This observation leads to a sequence of works, including TiDE [5], MTS-Mixers [28], TSMixer [28], TimeMixer [48], HDMixer [16], SOFTS [12], FITS [53], SparseTSF [32], and SSCNN [6]. The proposed CycleNet in this paper is also a Linear- or MLP-based model that is simple, efficient, and powerful for time series forecasting.

**RNN-based Models**    Conceptually, Recurrent Neural Networks (RNN) are considered to be the most suitable models for modeling time series data [24, 43]. However, due to difficulties in parallelization and modeling long sequences, RNNs are not the most popular choice in works on LTSF tasks. Recent works aim to revitalize RNN models in long sequence modeling tasks, such as SegRNN [31], WITRAN [19], SutraNets [2], and RWKV-TS [15].

**TCN-based Models**    Because of the parallelizability of convolution operations and their ability to capture features at different time scales, Temporal Convolutional Networks (TCN) methods are considered strong competitors for addressing time series tasks [1, 9]. Recent works that apply TCN methods to LTSF tasks include SCINet [34], MICN [47], TimesNet [52], PatchMixer [10], and ModernTCN [38].

**LLM-based Models**    The remarkable capabilities demonstrated by large language models (LLM) have sparked interest among researchers from various fields, including those working on time series forecasting tasks [21, 18]. Some works consider fine-tuning pre-trained LLMs to perform time series analysis tasks, including OFA [61], Time-LLM [20], and TEMPO [3]. Other works aim to achieve zero-shot inference using large pre-trained LLMs through prompt engineering, including LLMTime [11], PromptCast [54], and LSTPrompt [33].

# B  More details of CycleNet

## B.1  Overall pseudocode

Algorithm 1 demonstrates the implementation of modeling periodic patterns through recurrent cycles. Specifically, the first line defines the learnable parameter queue $Q$ and initializes it to zero. Lines 2-11 define the getCycle function, which will be called by CycleNet to obtain the corresponding truncated equivalent cyclic subsequences. This function takes two parameters, $i$ and $l$, where $i$ represents the relative positional index for $Q$, and $l$ represents the length of the required subsequence. $Q$ learns internal periodic patterns within the sequence through co-training with the backbone.

Furthermore, Algorithm 2 illustrates the workflow of CycleNet. The first step is to normalize the samples based on their mean and standard deviation, then call the getCycle function to remove the cyclic components of the input data. Subsequently, predict the residual components through the backbone. Finally, add back the cyclic components of the output data, and perform instance denormalization to obtain the final prediction result. Here, the cycle index $i$ corresponds to $t \mod W$, as described in Section 3.1.

**Algorithm 1** Modeling periodic patterns through recurrent cycles

---

**Require:** Number of channels $D$ and cycle length $W$
**Ensure:** Learned periodic patterns $Q \in \mathbb{R}^{W \times D}$
 1: Initialize learnable parameters $Q \leftarrow 0$                                     $\triangleright\ Q \in \mathbb{R}^{W \times D}$
 2: **function** GETCYCLE$(i, l)$                                         $\triangleright$ Define function
 3:     $Q' \leftarrow \mathrm{Roll}(Q, \mathrm{shifts} = -i, \mathrm{dim} = 0)$         $\triangleright$ Roll the queue to the appropriate index
 4:     **if** $l < W$ **then**                        $\triangleright$ Retrieve the required part directly from $Q'$
 5:         **return** $Q'_{0:l}$
 6:     **else**                                  $\triangleright$ Repeat $Q'$ to match the required length
 7:         $n \leftarrow \lfloor l/W \rfloor$
 8:         $d \leftarrow l \mod W$
 9:         **return** $\mathrm{Concat}([Q'] \times n, [Q'_{0:d}])$     $\triangleright$ Concatenate replicated $Q'$ and the remaining part
10:     **end if**
11: **end function**

---

**Algorithm 2** Workflow of CycleNet

---

**Require:** Look-back length $L$, forecast horizon $H$, cycle index $i$, and input $x_{t-L+1:t} \in \mathbb{R}^{L \times D}$
**Ensure:** Forecast output $\bar{x}_{t+1:t+H} \in \mathbb{R}^{H \times D}$
 1: **if** RevIN is applied **then**
 2:     $\mu, \sigma \leftarrow \mathrm{Mean}(x_{t-L+1:t}), \mathrm{STD}(x_{t-L+1:t})$        $\triangleright$ Compute mean and standard deviation
 3:     $x_{t-L+1:t} \leftarrow \frac{x_{t-L+1:t} - \mu}{\sigma + \epsilon}$                 $\triangleright$ Remove instance-specific statistics
 4: **end if**
 5: $x'_{t-L+1:t} \leftarrow x_{t-L+1:t} - \mathrm{getCycle}(i, L)$           $\triangleright$ Remove the cycle component
 6: $\bar{x}'_{t+1:t+H} \leftarrow \mathrm{Backbone}(x'_{t-L+1:t})$             $\triangleright$ Forecast using backbone model
 7: $\bar{x}_{t+1:t+H} \leftarrow \bar{x}'_{t+1:t+H} + \mathrm{getCycle}(i + L, H)$      $\triangleright$ Restore the cycle component
 8: **if** RevIN is applied **then**
 9:     $\bar{x}_{t+1:t+H} \leftarrow \bar{x}_{t+1:t+H} \times (\sigma + \epsilon) + \mu$          $\triangleright$ Restore instance-specific statistics
10: **end if**

---

### B.2 Utilizing ACF analysis to determine cycle length

The RCF technique utilizes recurrent cycles $Q \in \mathbb{R}^{W \times D}$ to model the internal periodic patterns of sequences. Here, the hyperparameter $W$ determines the length of the recurrent cycles, which should precisely match the length of the periodic patterns within the data. As shown in the results of Table 6, when $W$ is not accurately set, the RCF technique fails to fulfill its intended purpose. Although, in practice, we can infer the maximum cycle length of the dataset by considering the data's sampling frequency and the potential existing periodic patterns (as shown in Table 1), this manual inference method may introduce errors. Therefore, we may need a more scientific and precise approach to find the hyperparameter $W$.

In such cases, the autocorrelation function (ACF) [39] serves as a powerful mathematical tool to help us determine the periodicity within the data. The autocorrelation function measures the correlation between a time series and its lagged values, indicating the presence of autocorrelation within the data. Mathematically, this can be expressed as:

$$ACF = \frac{\sum_{t=1}^{N-k}(x_t - \bar{x})(x_{t+k} - \bar{x})}{\sum_{t=1}^{N}(x_t - \bar{x})^2}, \tag{6}$$

where $N$ represents the total number of observations, $x_t$ denotes the value of the time series at time $t$, $k$ is the lag time, and $\bar{x}$ is the mean of the time series values.

Here, when the lag time $k$ aligns with the data's cycle, the ACF value exhibits a significant peak. Specifically, the largest peak corresponds to the lag that aligns with the length of the maximum cycle present in the dataset. Conversely, if the data lacks periodicity, no significant peaks or troughs will be observed.

We present the ACF results for each dataset in Figure 6. It can be observed that these datasets all display evident periodicity, indicated by prominent peaks and troughs in the plots. More importantly,

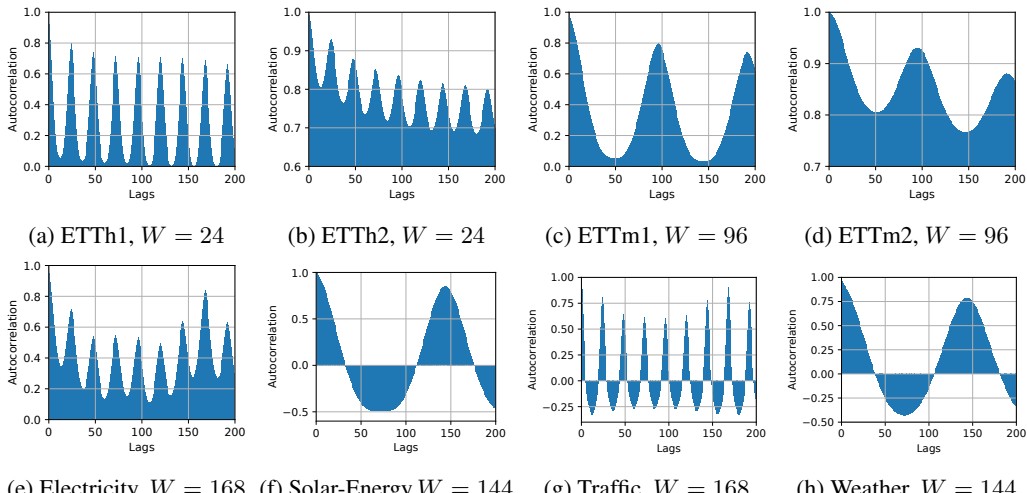

(a) ETTh1, $W = 24$      (b) ETTh2, $W = 24$      (c) ETTm1, $W = 96$      (d) ETTm2, $W = 96$

(e) Electricity, $W = 168$   (f) Solar-Energy,$W = 144$   (g) Traffic, $W = 168$    (h) Weather, $W = 144$

Figure 6: Visualization of ACF results on the training set of different datasets. The hyperparameter $W$ should be set to the lag corresponding to the observed maximum peak.

the maximum cycles shown in the plots align with the pre-inferred cycle lengths from Table 1. This indicates the correctness of the pre-inferred lengths, and $W$ should be strictly set to these values.

### B.3 Experimental details

We utilized widely used benchmark datasets for LTSF tasks, including the ETT series, Electricity, Solar-Energy, Traffic, and Weather. Following prior works such as Autoformer [51] and iTransformer [37], we split the ETTs dataset into training, validation, and test sets with a ratio of 6:2:2, while the other datasets were split in a ratio of 7:1:2.

We implemented CycleNet using PyTorch [41] and conducted experiments on a single NVIDIA RTX 4090 GPU with 24GB of memory. CycleNet was trained for 30 epochs with early stopping based on a patience of 5 on the validation set. The batch size was set uniformly to 256 for ETTs and the Weather dataset, and 64 for the remaining datasets. This adjustment was made because the latter datasets have a larger number of channels, requiring a relatively smaller batch size to avoid out-of-memory issues. The learning rate was selected from the range {0.002, 0.005, 0.01} based on the performance on the validation set. The hyperparameter $W$ was set consistently to the pre-inferred cycle length as shown in Table 1. Additionally, the hidden layer size of CycleNet/MLP was uniformly set to 512.

By default, CycleNet uses RevIN without learnable affine parameters [22]. However, we found that on the Solar dataset, using RevIN leads to a significant performance drop, as shown in Appendix C.4. The primary reason for this may be that photovoltaic power generation data contains continuous segments of zero values (no power generation at night). When the look-back windows are not an integer multiple of a day, the calculation of means in RevIN can be significantly affected, leading to decreased performance. Therefore, for this dataset, we did not apply the RevIN strategy.

## C More experimental results

### C.1 Periodic patterns learned under different configurations

The proposed RCF technique can effectively learn the inherent periodic patterns within time series data. This capability is a significant advantage, revealing the potential value of RCF or its underlying cyclic modeling approach as a superior method to assist data engineers in analyzing patterns in time series data. To further elucidate the working principle behind RCF, we delve into the periodic patterns learned by the RCF technique under different configurations, as illustrated in Figure 7:

- **Forecast horizon** $H$: The learned patterns remain almost unchanged as the horizon length varies. This indicates that the horizon length does not affect the learned pattern results.

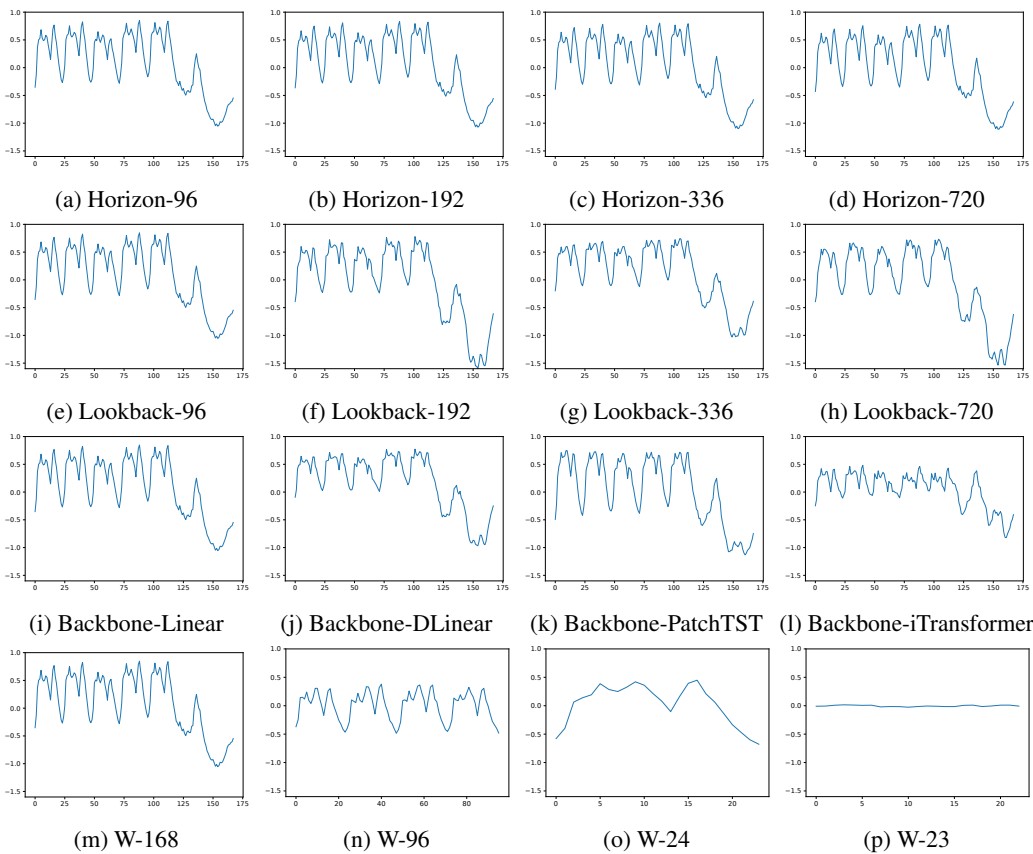

Figure 7: Periodic patterns of the 321st channel in the Electricity dataset, learned under different configurations. The basic configuration includes both a look-back and horizon length of 96, a simple Linear model as the backbone, and the correct cycle length $W$ set to 168.

- **Look-back length** $L$: The overall pattern remains unchanged as the look-back window changes. However, with closer observation, it is noticeable that the learned pattern becomes smoother with an increased look-back. This is because a longer look-back provides the backbone with richer periodic information, thereby reducing the reliance on the learned pattern component.

- **Backbone**: The patterns vary somewhat with different backbones. When DLinear is used as the backbone, the learned patterns are smoother, as DLinear's decomposition technique itself extracts certain periodic features. When iTransformer is the backbone, the learned patterns differ more, as it additionally models multichannel relationships, so the learned periodic patterns may consider multichannel feature interactions. PatchTST's performance is more similar to that of Linear, as it is also a regular single-channel modeling method, though with stronger nonlinear learning capabilities compared to the Linear model.

- **Cycle length** $W$: When $W$ is set to 168 (the weekly cycle length for the Electricity dataset), the recurrent cycle $Q$ learns the complete periodic pattern, including both weekly and daily cycles. When $W$ is set to 24 (the daily cycle length), the recurrent cycle $Q$ only learns the daily cycle pattern. When $W$ is set to 96 (four times the daily cycle length), the recurrent cycle $Q$ learns four repeated daily cycle models. However, when $W$ is set to 23 (without matching any semantic meaning), the recurrent cycle $Q$ fails to learn any meaningful pattern, resulting in a straight line.

## C.2 Full results with different look-back lengths

Table 2 presents the comparison results of CycleNet with other models on the mean performance at look-back length $L = 96$ for various forecast horizons $H \in \{96, 192, 336, 720\}$. Here, we further

Table 7: Full results of different models with the look-back length $L = 96$. The reported results with standard deviation of CycleNet are averaged from 5 runs (with different random seeds of $\{2024, 2025, 2026, 2027, 2028\}$). The results of other models are sourced from iTransformer [37]. The best results are highlighted in **bold** and the second best are underlined.

| Model | | FEDformer | | TimesNet | | iTransformer | | CycleNet/Linear | | CycleNet/MLP | |
|---|---|---|---|---|---|---|---|---|---|---|---|
| Metric | | MSE | MAE | MSE | MAE | MSE | MAE | MSE | MAE | MSE | MAE |
| ETTh1 | 96 | 0.376 | 0.419 | 0.384 | 0.402 | 0.386 | 0.405 | 0.378±0.001 | **0.391**±0.001 | **0.375**±0.001 | 0.395±0.001 |
| | 192 | **0.420** | 0.448 | 0.436 | 0.429 | 0.441 | 0.436 | 0.426±0.001 | **0.419**±0.001 | 0.436±0.002 | 0.428±0.002 |
| | 336 | **0.459** | 0.465 | 0.491 | 0.469 | 0.487 | 0.458 | 0.464±0.001 | **0.439**±0.001 | 0.496±0.001 | 0.455±0.003 |
| | 720 | 0.506 | 0.507 | 0.521 | 0.500 | 0.503 | 0.491 | **0.461**±0.001 | **0.460**±0.001 | 0.520±0.021 | 0.484±0.012 |
| | Avg | 0.440 | 0.460 | 0.458 | 0.450 | 0.454 | 0.448 | **0.432**±0.001 | **0.427**±0.001 | 0.457±0.006 | 0.441±0.004 |
| ETTh2 | 96 | 0.358 | 0.397 | 0.340 | 0.374 | 0.297 | 0.349 | **0.285**±0.001 | **0.335**±0.001 | 0.298±0.003 | 0.344±0.001 |
| | 192 | 0.429 | 0.439 | 0.402 | 0.414 | 0.380 | 0.400 | 0.373±0.001 | 0.391±0.001 | 0.372±0.002 | 0.396±0.002 |
| | 336 | 0.496 | 0.487 | 0.452 | 0.452 | 0.428 | **0.432** | **0.421**±0.001 | 0.433±0.001 | 0.431±0.007 | 0.439±0.005 |
| | 720 | 0.463 | 0.474 | 0.462 | 0.468 | **0.427** | **0.445** | 0.453±0.003 | 0.458±0.002 | 0.450±0.010 | 0.458±0.005 |
| | Avg | 0.437 | 0.449 | 0.414 | 0.427 | **0.383** | 0.407 | **0.383**±0.001 | **0.404**±0.001 | 0.388±0.005 | 0.409±0.003 |
| ETTm1 | 96 | 0.379 | 0.419 | 0.338 | 0.375 | 0.334 | 0.368 | 0.325±0.001 | 0.363±0.001 | 0.319±0.001 | 0.360±0.001 |
| | 192 | 0.426 | 0.441 | 0.374 | 0.387 | 0.377 | 0.391 | 0.366±0.001 | 0.382±0.001 | **0.360**±0.002 | **0.381**±0.001 |
| | 336 | 0.445 | 0.459 | 0.410 | 0.411 | 0.426 | 0.420 | 0.396±0.001 | **0.401**±0.001 | **0.389**±0.001 | 0.403±0.001 |
| | 720 | 0.543 | 0.490 | 0.478 | 0.450 | 0.491 | 0.459 | 0.457±0.001 | **0.433**±0.001 | 0.447±0.001 | 0.441±0.001 |
| | Avg | 0.448 | 0.452 | 0.400 | 0.406 | 0.407 | 0.410 | 0.386±0.001 | **0.395**±0.001 | **0.379**±0.001 | 0.396±0.001 |
| ETTm2 | 96 | 0.203 | 0.287 | 0.187 | 0.267 | 0.180 | 0.264 | 0.166±0.001 | 0.248±0.001 | **0.163**±0.001 | **0.246**±0.001 |
| | 192 | 0.269 | 0.328 | 0.249 | 0.309 | 0.250 | 0.309 | 0.233±0.001 | 0.291±0.001 | **0.229**±0.001 | **0.290**±0.001 |
| | 336 | 0.325 | 0.366 | 0.321 | 0.351 | 0.311 | 0.348 | 0.293±0.001 | 0.330±0.001 | **0.284**±0.001 | **0.327**±0.001 |
| | 720 | 0.421 | 0.415 | 0.408 | 0.403 | 0.412 | 0.407 | 0.395±0.001 | 0.389±0.001 | **0.389**±0.003 | 0.391±0.002 |
| | Avg | 0.305 | 0.349 | 0.291 | 0.333 | 0.288 | 0.332 | 0.272±0.001 | 0.315±0.001 | **0.266**±0.001 | **0.314**±0.001 |
| Electricity | 96 | 0.193 | 0.308 | 0.168 | 0.272 | 0.148 | 0.240 | 0.141±0.001 | 0.234±0.001 | **0.136**±0.001 | **0.229**±0.001 |
| | 192 | 0.201 | 0.315 | 0.184 | 0.289 | 0.162 | 0.253 | 0.155±0.001 | 0.247±0.001 | **0.152**±0.001 | **0.244**±0.001 |
| | 336 | 0.214 | 0.329 | 0.198 | 0.300 | 0.178 | 0.269 | 0.172±0.001 | **0.264**±0.001 | **0.170**±0.001 | **0.264**±0.001 |
| | 720 | 0.246 | 0.355 | 0.220 | 0.320 | 0.225 | 0.317 | **0.210**±0.001 | **0.296**±0.001 | 0.212±0.001 | 0.299±0.001 |
| | Avg | 0.214 | 0.327 | 0.193 | 0.295 | 0.178 | 0.270 | 0.170±0.001 | 0.260±0.001 | **0.168**±0.001 | **0.259**±0.001 |
| Solar-Energy | 96 | 0.242 | 0.342 | 0.250 | 0.292 | 0.203 | **0.237** | 0.209±0.001 | 0.260±0.003 | **0.190**±0.007 | 0.247±0.003 |
| | 192 | 0.285 | 0.380 | 0.296 | 0.318 | 0.233 | **0.261** | 0.231±0.002 | 0.269±0.002 | **0.210**±0.004 | 0.266±0.008 |
| | 336 | 0.282 | 0.376 | 0.319 | 0.330 | 0.248 | 0.273 | 0.246±0.002 | 0.275±0.003 | **0.217**±0.006 | **0.266**±0.006 |
| | 720 | 0.357 | 0.427 | 0.338 | 0.337 | 0.249 | 0.275 | 0.255±0.001 | 0.274±0.003 | **0.223**±0.003 | **0.266**±0.003 |
| | Avg | 0.292 | 0.381 | 0.301 | 0.319 | 0.233 | 0.262 | 0.235±0.001 | 0.270±0.002 | **0.210**±0.005 | **0.261**±0.005 |
| Traffic | 96 | 0.587 | 0.366 | 0.593 | 0.321 | **0.395** | **0.268** | 0.480±0.001 | 0.314±0.001 | 0.458±0.001 | 0.296±0.001 |
| | 192 | 0.604 | 0.373 | 0.617 | 0.336 | **0.417** | **0.276** | 0.482±0.001 | 0.313±0.001 | 0.457±0.001 | 0.294±0.001 |
| | 336 | 0.621 | 0.383 | 0.629 | 0.336 | **0.433** | **0.283** | 0.476±0.001 | 0.303±0.001 | 0.470±0.001 | 0.299±0.001 |
| | 720 | 0.626 | 0.382 | 0.640 | 0.350 | **0.467** | **0.302** | 0.503±0.001 | 0.320±0.001 | 0.502±0.001 | 0.314±0.001 |
| | Avg | 0.610 | 0.376 | 0.620 | 0.336 | **0.428** | **0.282** | 0.485±0.001 | 0.313±0.001 | 0.472±0.001 | 0.301±0.001 |
| Weather | 96 | 0.217 | 0.296 | 0.172 | 0.220 | 0.174 | 0.214 | 0.170±0.001 | 0.216±0.001 | **0.158**±0.001 | **0.203**±0.001 |
| | 192 | 0.276 | 0.336 | 0.219 | 0.261 | 0.221 | 0.254 | 0.222±0.001 | 0.259±0.001 | **0.207**±0.001 | **0.247**±0.001 |
| | 336 | 0.339 | 0.380 | 0.280 | 0.306 | 0.278 | 0.296 | 0.275±0.001 | 0.296±0.001 | **0.262**±0.001 | **0.289**±0.001 |
| | 720 | 0.403 | 0.428 | 0.365 | 0.359 | 0.358 | 0.349 | 0.349±0.001 | 0.345±0.001 | **0.344**±0.001 | **0.344**±0.001 |
| | Avg | 0.309 | 0.360 | 0.259 | 0.287 | 0.258 | 0.278 | 0.254±0.001 | 0.279±0.001 | **0.243**±0.001 | **0.271**±0.001 |

showcase the complete comparison results for different forecast horizons in Table 7. It can be observed that in most settings, CycleNet achieves state-of-the-art results, consistent with the findings in Table 2. Additionally, the standard deviation of CycleNet's results is mostly below 0.001. This strongly indicates the robustness of CycleNet.

Additionally, the look-back length is a crucial hyperparameter that significantly impacts the performance of time series forecasting models, as it determines the richness of information the model can leverage. Initially, the community focused primarily on exploring the application of Transformers in time series forecasting tasks. Due to the inherent complexity of Transformers, using excessively long look-back windows resulted in a significant increase in runtime. As a result, many popular models at the time, such as Informer [59], Autoformer [51], and FEDformer [60], employed shorter look-back windows, typically with $L = 96$.

Table 8: Full results of different models with longer look-back lengths $L \in \{336, 720\}$. The reported results of CycleNet are averaged from 5 runs (with different random seeds of $\{2024, 2025, 2026, 2027, 2028\}$). The results of other models are *reproduced* after fixing a long-standing bug (discarding the last batch of data during the test phase). The best results are highlighted in **bold** and the second best are underlined.

| Lookback | | L=336 DLinear [2023] MSE | MAE | PatchTST [2023] MSE | MAE | CycleNet /Linear MSE | MAE | CycleNet /MLP MSE | MAE | L=720 SegRNN [2023] MSE | MAE | SparseTSF [2024] MSE | MAE | CycleNet /Linear MSE | MAE | CycleNet /MLP MSE | MAE |
|---|---|---|---|---|---|---|---|---|---|---|---|---|---|---|---|---|---|
| ETTh1 | 96 | **0.374** | 0.398 | 0.385 | 0.405 | **0.374** | **0.396** | 0.382 | 0.403 | **0.351** | 0.392 | 0.362 | **0.388** | 0.379 | 0.403 | 0.385 | 0.412 |
| | 192 | 0.430 | 0.440 | 0.414 | 0.421 | **0.406** | **0.415** | 0.421 | 0.426 | **0.390** | 0.418 | 0.403 | **0.411** | 0.416 | 0.425 | 0.424 | 0.438 |
| | 336 | 0.442 | 0.445 | 0.440 | 0.440 | **0.431** | **0.430** | 0.449 | 0.444 | 0.449 | 0.452 | **0.434** | **0.428** | 0.447 | 0.445 | 0.460 | 0.463 |
| | 720 | 0.497 | 0.507 | 0.456 | 0.470 | **0.450** | 0.464 | 0.497 | 0.485 | 0.492 | 0.494 | **0.426** | 0.447 | 0.477 | 0.483 | 0.486 | 0.487 |
| | Avg | 0.436 | 0.448 | 0.424 | 0.434 | **0.415** | **0.426** | 0.437 | 0.440 | 0.421 | 0.439 | **0.406** | **0.419** | 0.430 | 0.439 | 0.439 | 0.450 |
| ETTh2 | 96 | 0.281 | 0.347 | **0.275** | **0.337** | 0.279 | 0.341 | 0.300 | 0.355 | 0.275 | 0.338 | 0.294 | 0.346 | **0.271** | **0.337** | 0.293 | 0.352 |
| | 192 | 0.367 | 0.404 | **0.338** | **0.379** | 0.342 | 0.385 | 0.373 | 0.403 | 0.338 | 0.380 | 0.339 | **0.377** | **0.332** | 0.380 | 0.359 | 0.395 |
| | 336 | 0.438 | 0.454 | **0.365** | **0.398** | 0.371 | 0.413 | 0.384 | 0.419 | 0.419 | 0.445 | **0.359** | **0.397** | 0.362 | 0.408 | 0.392 | 0.423 |
| | 720 | 0.598 | 0.549 | **0.391** | **0.429** | 0.426 | 0.451 | 0.428 | 0.450 | 0.431 | 0.464 | **0.383** | **0.424** | 0.415 | 0.449 | 0.425 | 0.451 |
| | Avg | 0.421 | 0.439 | **0.342** | **0.386** | 0.355 | 0.398 | 0.371 | 0.407 | 0.366 | 0.407 | **0.344** | **0.386** | 0.345 | 0.394 | 0.367 | 0.405 |
| ETTm1 | 96 | 0.307 | 0.350 | **0.291** | **0.343** | 0.299 | 0.348 | 0.297 | 0.351 | **0.295** | 0.356 | 0.312 | 0.354 | 0.307 | **0.353** | 0.301 | 0.357 |
| | 192 | 0.340 | 0.373 | **0.334** | 0.370 | **0.334** | **0.367** | 0.338 | 0.377 | **0.334** | 0.382 | 0.347 | 0.376 | 0.337 | **0.371** | 0.341 | 0.377 |
| | 336 | 0.377 | 0.397 | **0.367** | 0.392 | 0.368 | **0.386** | 0.374 | 0.400 | **0.359** | 0.401 | 0.367 | **0.386** | 0.364 | 0.387 | 0.376 | 0.396 |
| | 720 | 0.433 | 0.433 | 0.422 | 0.426 | **0.417** | **0.414** | 0.436 | 0.431 | 0.415 | 0.435 | 0.419 | 0.413 | **0.410** | **0.411** | 0.431 | 0.425 |
| | Avg | 0.364 | 0.388 | **0.354** | 0.383 | 0.355 | **0.379** | 0.361 | 0.390 | **0.351** | 0.394 | 0.361 | 0.382 | 0.355 | **0.381** | 0.362 | 0.389 |
| ETTm2 | 96 | 0.165 | 0.257 | 0.164 | 0.254 | **0.159** | **0.247** | 0.178 | 0.262 | 0.165 | 0.251 | 0.163 | 0.252 | **0.159** | **0.249** | 0.176 | 0.265 |
| | 192 | 0.227 | 0.307 | 0.221 | 0.293 | **0.214** | **0.286** | 0.238 | 0.303 | 0.226 | 0.300 | 0.217 | 0.290 | **0.214** | **0.289** | 0.231 | 0.305 |
| | 336 | 0.304 | 0.362 | 0.276 | 0.328 | **0.269** | **0.322** | 0.292 | 0.339 | 0.282 | 0.341 | 0.270 | 0.327 | **0.268** | **0.326** | 0.282 | 0.338 |
| | 720 | 0.431 | 0.441 | 0.366 | 0.383 | **0.363** | **0.382** | 0.374 | 0.391 | 0.361 | 0.392 | **0.352** | **0.379** | 0.353 | 0.384 | 0.361 | 0.388 |
| | Avg | 0.282 | 0.342 | 0.257 | 0.315 | **0.251** | **0.309** | 0.271 | 0.324 | 0.259 | 0.321 | 0.251 | **0.312** | **0.249** | 0.312 | 0.263 | 0.324 |
| Electricity | 96 | 0.140 | 0.237 | 0.131 | 0.225 | 0.128 | 0.223 | **0.126** | **0.221** | 0.130 | 0.228 | 0.138 | 0.233 | 0.128 | **0.223** | **0.127** | **0.223** |
| | 192 | 0.153 | 0.250 | 0.148 | 0.240 | **0.144** | **0.237** | **0.144** | **0.237** | 0.152 | 0.251 | 0.151 | 0.244 | **0.143** | **0.237** | 0.144 | 0.239 |
| | 336 | 0.169 | 0.267 | 0.165 | 0.259 | **0.160** | **0.254** | **0.160** | 0.255 | 0.170 | 0.272 | 0.166 | 0.260 | **0.159** | **0.254** | **0.159** | 0.255 |
| | 720 | 0.203 | 0.299 | 0.202 | 0.291 | **0.198** | **0.287** | 0.199 | 0.291 | 0.203 | 0.304 | 0.205 | 0.293 | 0.197 | **0.287** | **0.196** | 0.290 |
| | Avg | 0.166 | 0.263 | 0.162 | 0.254 | 0.158 | **0.250** | **0.157** | 0.251 | 0.164 | 0.264 | 0.165 | 0.258 | **0.157** | **0.250** | **0.157** | 0.252 |
| Solar-Energy | 96 | 0.222 | 0.292 | 0.190 | 0.278 | 0.200 | 0.250 | **0.182** | **0.245** | 0.175 | 0.236 | 0.195 | 0.243 | 0.194 | 0.255 | **0.174** | **0.232** |
| | 192 | 0.249 | 0.313 | 0.206 | **0.252** | 0.221 | 0.261 | **0.191** | 0.254 | 0.193 | 0.268 | 0.215 | 0.254 | 0.205 | 0.251 | **0.187** | **0.246** |
| | 336 | 0.268 | 0.327 | 0.217 | **0.254** | 0.236 | 0.272 | **0.197** | 0.257 | 0.209 | 0.263 | 0.232 | 0.262 | 0.218 | 0.257 | **0.194** | **0.252** |
| | 720 | 0.271 | 0.326 | 0.219 | **0.255** | 0.245 | 0.277 | **0.207** | 0.264 | 0.205 | 0.264 | 0.237 | 0.263 | 0.239 | 0.278 | **0.201** | **0.259** |
| | Avg | 0.253 | 0.315 | 0.208 | 0.260 | 0.226 | 0.265 | **0.194** | **0.255** | 0.196 | 0.258 | 0.220 | 0.256 | 0.214 | 0.260 | **0.189** | **0.247** |
| Traffic | 96 | 0.410 | 0.282 | **0.373** | **0.254** | 0.397 | 0.278 | 0.386 | 0.268 | **0.356** | **0.255** | 0.389 | 0.268 | 0.381 | 0.266 | 0.374 | 0.268 |
| | 192 | 0.423 | 0.288 | **0.391** | **0.262** | 0.411 | 0.283 | 0.404 | 0.276 | **0.374** | **0.268** | 0.398 | 0.270 | 0.394 | 0.273 | 0.390 | 0.275 |
| | 336 | 0.436 | 0.296 | **0.404** | **0.269** | 0.424 | 0.289 | 0.416 | 0.281 | **0.393** | **0.273** | 0.411 | 0.275 | 0.406 | 0.279 | 0.405 | 0.282 |
| | 720 | 0.466 | 0.315 | **0.436** | **0.287** | 0.450 | 0.305 | 0.445 | 0.300 | **0.434** | **0.294** | 0.448 | 0.297 | 0.441 | 0.300 | 0.441 | 0.302 |
| | Avg | 0.434 | 0.295 | **0.401** | **0.268** | 0.421 | 0.289 | 0.413 | 0.281 | **0.389** | **0.273** | 0.412 | 0.278 | 0.406 | 0.280 | 0.403 | 0.282 |
| Weather | 96 | 0.174 | 0.235 | 0.155 | 0.204 | 0.167 | 0.221 | **0.148** | **0.200** | **0.141** | 0.205 | 0.169 | 0.223 | 0.164 | 0.220 | 0.149 | **0.203** |
| | 192 | 0.219 | 0.281 | 0.195 | 0.242 | 0.212 | 0.258 | **0.190** | **0.240** | **0.185** | 0.250 | 0.214 | 0.262 | 0.209 | 0.258 | 0.192 | **0.244** |
| | 336 | 0.264 | 0.317 | 0.249 | **0.283** | 0.260 | 0.293 | **0.243** | **0.283** | **0.241** | 0.297 | 0.257 | 0.293 | 0.255 | 0.292 | 0.242 | **0.283** |
| | 720 | 0.324 | 0.363 | **0.321** | **0.334** | 0.328 | 0.339 | 0.322 | 0.339 | 0.318 | 0.352 | 0.321 | 0.340 | 0.320 | 0.338 | **0.312** | **0.333** |
| | Avg | 0.245 | 0.299 | 0.230 | **0.266** | 0.242 | 0.278 | **0.226** | **0.266** | **0.221** | 0.276 | 0.240 | 0.280 | 0.237 | 0.277 | 0.224 | **0.266** |

With the recent development of model lightweighting techniques, particularly the adoption of channel-independent strategies (first applied in DLinear [56] and PatchTST [40]), more models have started to experiment with longer look-back windows in pursuit of higher predictive accuracy. For instance, DLinear and PatchTST default to using look-back windows of $L = 336$, while SegRNN [31] and SparseTSF [32] default to using $L = 720$. To explore CycleNet's performance with longer look-back windows, we compared CycleNet with these advanced models using their respective default, longer look-back windows in Table 8.

It is important to note that we re-ran the official open-source code of these baselines to obtain the corresponding results, using the same MSE as the loss function (as SegRNN originally used MAE as its loss). Additionally, there was a long-standing bug in their original repositories, where the data from the last batch was discarded during testing [42, 53]. This issue could have affected the model's performance, so we fixed this problem before re-running the experiments.

It can be observed that even with a longer look-back length, CycleNet generally maintains a significant advantage, achieving state-of-the-art performance in most scenarios. This demonstrates CycleNet's excellent performance across different look-back lengths. It is worth noting that both PatchTST and

SegRNN outperform CycleNet on the Traffic dataset, even though they are also channel-independent models. This is partly because the Traffic dataset contains more outliers (see more discussion in Appendix C.5), which may impact the performance of RCF; additionally, PatchTST and SegRNN are more complex deep models with stronger nonlinear capabilities, enabling them to fit various patterns across numerous channels (the Traffic dataset has up to 862 channels).

## C.3 Full results with different STD techniques

Table 9: Full results of comparison of different STD techniques. The configuration used here is consistent with that of DLinear [56], where a pure Linear model serves as the backbone, a look-back length of 336 is employed, and no additional instance normalization strategies are applied. Thus, CLinear here refers to CycleNet/Linear without RevIN. The best results are highlighted in **bold** and the second best are underlined.

| Model | | CLinear (RCF+Linear) | | LDLinear (LD+Linear) | | DLinear (MOV+Linear) | | SLinear (Sparse+Linear) | | Linear | |
|---|---|---|---|---|---|---|---|---|---|---|---|
| Metric | | MSE | MAE | MSE | MAE | MSE | MAE | MSE | MAE | MSE | MAE |
| ETTh1 | 96 | 0.370 | 0.395 | 0.372 | 0.394 | 0.372 | 0.394 | **0.366** | **0.388** | 0.374 | 0.395 |
| | 192 | **0.404** | 0.417 | 0.410 | 0.420 | 0.408 | 0.417 | 0.406 | **0.414** | 0.409 | 0.418 |
| | 336 | **0.434** | **0.440** | 0.449 | 0.452 | 0.441 | 0.442 | 0.440 | 0.442 | 0.442 | 0.444 |
| | 720 | **0.465** | **0.486** | 0.476 | 0.492 | 0.480 | 0.494 | 0.483 | 0.501 | 0.484 | 0.498 |
| | Avg | **0.418** | **0.434** | 0.427 | 0.439 | 0.425 | 0.437 | 0.424 | 0.436 | 0.427 | 0.439 |
| ETTh2 | 96 | 0.308 | 0.369 | **0.292** | **0.357** | 0.297 | 0.362 | 0.340 | 0.389 | 0.305 | 0.368 |
| | 192 | 0.382 | 0.416 | **0.372** | **0.409** | 0.398 | 0.426 | 0.379 | 0.413 | 0.385 | 0.419 |
| | 336 | 0.454 | 0.465 | 0.479 | 0.480 | 0.496 | 0.489 | **0.404** | **0.437** | 0.458 | 0.470 |
| | 720 | **0.661** | **0.575** | 0.675 | 0.582 | 0.694 | 0.592 | 0.720 | 0.600 | 0.691 | 0.592 |
| | Avg | **0.451** | **0.456** | 0.455 | 0.457 | 0.471 | 0.467 | 0.460 | 0.460 | 0.460 | 0.462 |
| ETTm1 | 96 | **0.298** | 0.350 | 0.305 | 0.350 | 0.309 | 0.356 | 0.306 | **0.349** | 0.305 | 0.349 |
| | 192 | **0.330** | 0.370 | 0.335 | **0.366** | 0.346 | 0.380 | 0.339 | 0.370 | 0.338 | 0.369 |
| | 336 | **0.359** | **0.388** | 0.372 | 0.390 | 0.373 | 0.391 | 0.372 | 0.389 | 0.371 | 0.389 |
| | 720 | **0.410** | **0.421** | 0.445 | 0.443 | 0.439 | 0.435 | 0.430 | 0.426 | 0.433 | 0.428 |
| | Avg | **0.349** | **0.382** | 0.365 | 0.387 | 0.367 | 0.390 | 0.362 | 0.383 | 0.362 | 0.384 |
| ETTm2 | 96 | **0.164** | 0.260 | 0.165 | **0.257** | 0.165 | 0.257 | 0.177 | 0.272 | 0.166 | 0.259 |
| | 192 | **0.225** | **0.304** | 0.240 | 0.318 | 0.232 | 0.310 | 0.246 | 0.325 | 0.228 | 0.305 |
| | 336 | **0.271** | **0.332** | 0.290 | 0.349 | 0.295 | 0.356 | 0.309 | 0.370 | 0.275 | 0.334 |
| | 720 | 0.406 | 0.423 | **0.396** | **0.419** | 0.427 | 0.442 | 0.427 | 0.440 | 0.407 | 0.425 |
| | Avg | **0.266** | **0.330** | 0.273 | 0.336 | 0.280 | 0.341 | 0.290 | 0.352 | 0.269 | 0.331 |
| Electricity | 96 | **0.131** | **0.228** | 0.140 | 0.237 | 0.140 | 0.237 | 0.148 | 0.243 | 0.140 | 0.238 |
| | 192 | **0.145** | **0.242** | 0.154 | 0.250 | 0.154 | 0.250 | 0.159 | 0.254 | 0.154 | 0.251 |
| | 336 | **0.160** | **0.260** | 0.170 | 0.268 | 0.169 | 0.268 | 0.173 | 0.271 | 0.170 | 0.269 |
| | 720 | **0.193** | **0.292** | 0.204 | 0.300 | 0.204 | 0.301 | 0.207 | 0.303 | 0.204 | 0.301 |
| | Avg | **0.157** | **0.255** | 0.167 | 0.264 | 0.167 | 0.264 | 0.172 | 0.268 | 0.167 | 0.265 |
| Solar-Energy | 96 | **0.192** | **0.251** | 0.222 | 0.294 | 0.222 | 0.298 | 0.226 | 0.296 | 0.224 | 0.302 |
| | 192 | **0.218** | **0.258** | 0.249 | 0.315 | 0.250 | 0.312 | 0.252 | 0.312 | 0.250 | 0.310 |
| | 336 | **0.231** | **0.262** | 0.268 | 0.326 | 0.270 | 0.335 | 0.270 | 0.326 | 0.269 | 0.325 |
| | 720 | **0.239** | **0.265** | 0.271 | 0.327 | 0.272 | 0.327 | 0.271 | 0.327 | 0.270 | 0.333 |
| | Avg | **0.220** | **0.259** | 0.253 | 0.316 | 0.254 | 0.318 | 0.255 | 0.315 | 0.253 | 0.318 |
| Traffic | 96 | **0.397** | **0.275** | 0.411 | 0.285 | 0.411 | 0.284 | 0.414 | 0.281 | 0.411 | 0.283 |
| | 192 | **0.412** | **0.282** | 0.423 | 0.288 | 0.423 | 0.289 | 0.425 | 0.285 | 0.423 | 0.289 |
| | 336 | **0.426** | **0.290** | 0.436 | 0.296 | 0.436 | 0.296 | 0.436 | 0.293 | 0.437 | 0.297 |
| | 720 | **0.456** | **0.308** | 0.466 | 0.315 | 0.466 | 0.316 | 0.464 | 0.310 | 0.466 | 0.316 |
| | Avg | **0.423** | **0.289** | 0.434 | 0.296 | 0.434 | 0.296 | 0.435 | 0.292 | 0.434 | 0.296 |
| Weather | 96 | 0.174 | 0.240 | **0.174** | **0.235** | 0.175 | 0.237 | 0.176 | 0.235 | 0.175 | 0.235 |
| | 192 | 0.218 | 0.279 | **0.215** | **0.271** | 0.215 | 0.273 | 0.218 | 0.277 | 0.218 | 0.276 |
| | 336 | 0.262 | 0.314 | 0.263 | 0.315 | **0.261** | **0.311** | 0.265 | 0.316 | 0.262 | 0.312 |
| | 720 | 0.328 | 0.367 | 0.325 | 0.365 | **0.324** | **0.363** | 0.325 | 0.363 | 0.327 | 0.366 |
| | Avg | 0.245 | 0.300 | 0.244 | 0.297 | **0.244** | **0.296** | 0.246 | 0.298 | 0.245 | 0.297 |

The proposed RCF technique is essentially a type of Seasonal-Trend Decomposition (STD) method. To directly compare RCF with existing related STD techniques, we adopted a strategy consistent with DLinear, using a pure Linear model as the backbone and not applying any instance normalization

techniques. We previously reported the mean performance of these techniques across different horizons $H \in \{96, 192, 336, 720\}$ in Table 5. Here, we further present the complete comparative results for all horizons in Table 9.

The results show that the RCF technique consistently outperforms other techniques. A notable exception is the relatively noisy weather dataset, where RCF does not show a significant advantage. However, in this case, the performance of several STD techniques is similar to that of the pure Linear model. Overall, these findings strongly support RCF as a new STD method that enhances model performance in scenarios with strong periodicity.

## C.4 Ablation study of RevIN

Table 10: Ablatioin results of RevIN.

| Model | CycleNet/L w. RevIN | | CycleNet/L w/o. RevIN | | RLinear [2023] | | CycleNet/M w. RevIN | | CycleNet/M w/o. RevIN | | RMLP [2023] | |
|---|---|---|---|---|---|---|---|---|---|---|---|---|
| Metric | MSE | MAE | MSE | MAE | MSE | MAE | MSE | MAE | MSE | MAE | MSE | MAE |
| **ETTh1** 96 | **0.377** | **0.391** | 0.379 | 0.399 | 0.385 | 0.393 | **0.378** | **0.397** | 0.383 | 0.401 | 0.383 | 0.401 |
| 192 | 0.426 | **0.419** | **0.423** | 0.428 | 0.439 | 0.424 | 0.440 | **0.431** | **0.431** | 0.436 | 0.437 | 0.432 |
| 336 | 0.464 | **0.439** | **0.460** | 0.452 | 0.483 | 0.448 | 0.495 | **0.453** | **0.486** | 0.467 | 0.494 | 0.461 |
| 720 | **0.462** | **0.460** | 0.484 | 0.494 | 0.481 | 0.470 | **0.502** | **0.473** | 0.547 | 0.516 | 0.540 | 0.499 |
| **ETTh2** 96 | **0.286** | **0.336** | 0.328 | 0.381 | 0.291 | 0.339 | **0.298** | **0.344** | 0.326 | 0.377 | 0.299 | 0.345 |
| 192 | **0.372** | 0.391 | 0.467 | 0.464 | 0.375 | **0.389** | 0.374 | 0.400 | 0.421 | 0.435 | **0.371** | **0.394** |
| 336 | 0.422 | 0.433 | 0.570 | 0.523 | **0.414** | **0.425** | 0.425 | 0.435 | 0.522 | 0.490 | **0.420** | **0.429** |
| 720 | 0.457 | 0.460 | 0.773 | 0.630 | **0.420** | **0.440** | 0.442 | 0.454 | 0.876 | 0.647 | **0.438** | **0.450** |
| **ETTm1** 96 | **0.325** | **0.363** | 0.327 | 0.371 | 0.351 | 0.372 | **0.320** | **0.361** | 0.338 | 0.383 | 0.327 | 0.366 |
| 192 | 0.366 | **0.382** | **0.359** | 0.388 | 0.390 | 0.390 | **0.361** | **0.382** | 0.367 | 0.393 | 0.370 | 0.386 |
| 336 | 0.396 | **0.402** | **0.391** | 0.414 | 0.423 | 0.414 | **0.392** | **0.404** | 0.396 | 0.419 | 0.404 | 0.410 |
| 720 | 0.457 | **0.434** | **0.434** | 0.442 | 0.486 | 0.448 | 0.448 | **0.441** | **0.447** | 0.448 | 0.462 | 0.445 |
| **ETTm2** 96 | **0.168** | **0.249** | 0.176 | 0.272 | 0.184 | 0.266 | **0.164** | **0.246** | 0.174 | 0.266 | 0.178 | 0.259 |
| 192 | **0.232** | **0.290** | 0.249 | 0.324 | 0.248 | 0.305 | **0.232** | **0.291** | 0.248 | 0.318 | 0.242 | 0.302 |
| 336 | **0.293** | **0.330** | 0.325 | 0.378 | 0.307 | 0.342 | **0.283** | **0.328** | 0.304 | 0.361 | 0.299 | 0.340 |
| 720 | **0.394** | **0.389** | 0.526 | 0.495 | 0.408 | 0.397 | **0.385** | **0.389** | 0.512 | 0.478 | 0.400 | 0.398 |
| **Electricity** 96 | **0.142** | **0.234** | 0.142 | 0.239 | 0.198 | 0.275 | **0.136** | **0.230** | 0.138 | 0.235 | 0.182 | 0.265 |
| 192 | 0.156 | **0.247** | **0.155** | 0.252 | 0.198 | 0.277 | **0.153** | **0.245** | 0.154 | 0.250 | 0.187 | 0.270 |
| 336 | 0.173 | **0.265** | **0.170** | 0.269 | 0.212 | 0.293 | **0.170** | **0.264** | 0.171 | 0.269 | 0.203 | 0.287 |
| 720 | 0.211 | **0.297** | **0.199** | 0.298 | 0.254 | 0.325 | 0.212 | **0.300** | **0.206** | 0.302 | 0.244 | 0.319 |
| **Solar** 96 | 0.250 | 0.277 | **0.208** | **0.256** | 0.308 | 0.332 | 0.195 | 0.252 | **0.187** | **0.245** | 0.236 | 0.270 |
| 192 | 0.289 | 0.299 | **0.231** | **0.269** | 0.345 | 0.349 | 0.225 | **0.272** | **0.215** | 0.275 | 0.270 | 0.290 |
| 336 | 0.338 | 0.323 | **0.247** | **0.272** | 0.387 | 0.364 | 0.248 | 0.289 | **0.212** | **0.257** | 0.296 | 0.305 |
| 720 | 0.351 | 0.326 | **0.258** | **0.275** | 0.390 | 0.358 | 0.253 | 0.286 | **0.228** | **0.269** | 0.296 | 0.303 |
| **Traffic** 96 | 0.480 | 0.314 | **0.475** | **0.302** | 0.647 | 0.386 | **0.459** | **0.297** | 0.469 | 0.298 | 0.510 | 0.331 |
| 192 | 0.482 | 0.313 | **0.475** | **0.305** | 0.600 | 0.362 | **0.457** | **0.295** | 0.477 | 0.304 | 0.505 | 0.327 |
| 336 | **0.476** | **0.303** | 0.489 | 0.313 | 0.607 | 0.365 | **0.470** | **0.300** | 0.487 | 0.302 | 0.518 | 0.332 |
| 720 | **0.505** | **0.321** | 0.518 | 0.327 | 0.644 | 0.383 | **0.502** | **0.314** | 0.522 | 0.315 | 0.553 | 0.350 |
| **Weather** 96 | **0.170** | **0.216** | 0.209 | 0.284 | 0.197 | 0.236 | **0.158** | **0.203** | 0.179 | 0.247 | 0.181 | 0.219 |
| 192 | **0.222** | **0.260** | 0.265 | 0.334 | 0.239 | 0.270 | **0.207** | **0.248** | 0.220 | 0.284 | 0.228 | 0.259 |
| 336 | **0.276** | **0.296** | 0.314 | 0.368 | 0.292 | 0.307 | **0.263** | **0.290** | 0.273 | 0.325 | 0.282 | 0.299 |
| 720 | **0.350** | **0.345** | 0.378 | 0.410 | 0.365 | 0.353 | **0.344** | **0.345** | 0.345 | 0.377 | 0.357 | 0.347 |

Instance normalization strategies constitute essential factors for the success of current models, such as PatchTST [40], TiDE [5], iTransformer [37], SparseTSF [32], etc. By default, CycleNet also adopts this strategy, namely the version of RevIN without learnable affine parameters [22]. Here, we meticulously investigate the impact of RevIN on the performance of CycleNet, and the results are shown in Table 10. On the ETTh2 and Weather datasets, RevIN significantly enhances the performance of CycleNet, possibly due to more severe distribution drift issues in these datasets. However, on the Solar dataset, RevIN leads to poorer performance, likely because the photovoltaic power generation data contains continuous segments of zero values (no power generation at night), which significantly affects the calculation of means in RevIN.

Overall, in most cases, RevIN leads to better performance. We acknowledge that RevIN is an indispensable cornerstone of CycleNet's success, but it is not the key factor that sets CycleNet apart from other models in terms of performance. As shown in the comparison results in Table 10, CycleNet exhibits a significant advantage over RLinear and RMLP, which can be viewed as CycleNet without RCF technique. This clearly demonstrates that the RCF technique is the key factor that significantly enhances the model's prediction accuracy, constituting the core contribution of this paper.

### C.5 Further Analysis in Traffic Scenarios

Table 11: Comparison results on the PEMS datasets. The look-back length $L$ is fixed at 96, and the forecast horizons are set to $H \in \{12, 24, 48, 96\}$. The results of other models are sourced from iTransformer [37]. The best results are highlighted in **bold**, and the second-best are underlined.

| Model | | CycleNet /MLP | | CycleNet /Linear | | RLinear [2023] | | iTransformer [2024] | | PatchTST [2023] | | Crossformer [2023] | | DLinear [2023] | | SCINet [2022] | |
|---|---|---|---|---|---|---|---|---|---|---|---|---|---|---|---|---|---|
| Metric | | MSE | MAE | MSE | MAE | MSE | MAE | MSE | MAE | MSE | MAE | MSE | MAE | MSE | MAE | MSE | MAE |
| PEMS03 | 12 | **0.066** | 0.172 | 0.080 | 0.192 | 0.126 | 0.236 | 0.071 | 0.174 | 0.099 | 0.216 | 0.090 | 0.203 | 0.122 | 0.243 | 0.066 | **0.172** |
| | 24 | 0.089 | 0.201 | 0.120 | 0.237 | 0.246 | 0.334 | 0.093 | 0.201 | 0.142 | 0.259 | 0.121 | 0.240 | 0.201 | 0.317 | **0.085** | **0.198** |
| | 48 | **0.136** | 0.247 | 0.156 | 0.258 | 0.551 | 0.529 | 0.125 | 0.236 | 0.211 | 0.319 | 0.202 | 0.317 | 0.333 | 0.425 | 0.127 | 0.238 |
| | 96 | 0.182 | 0.282 | 0.199 | 0.292 | 1.057 | 0.787 | **0.164** | **0.275** | 0.269 | 0.370 | 0.262 | 0.367 | 0.457 | 0.515 | 0.178 | 0.287 |
| PEMS04 | 12 | 0.078 | 0.186 | 0.089 | 0.201 | 0.138 | 0.252 | 0.078 | 0.183 | 0.105 | 0.224 | 0.098 | 0.218 | 0.148 | 0.272 | **0.073** | **0.177** |
| | 24 | 0.099 | 0.212 | 0.127 | 0.245 | 0.258 | 0.348 | 0.095 | 0.205 | 0.153 | 0.275 | 0.131 | 0.256 | 0.224 | 0.340 | **0.084** | **0.193** |
| | 48 | 0.133 | 0.248 | 0.169 | 0.286 | 0.572 | 0.544 | 0.120 | 0.233 | 0.229 | 0.339 | 0.205 | 0.326 | 0.355 | 0.437 | **0.099** | **0.211** |
| | 96 | 0.167 | 0.281 | 0.189 | 0.293 | 1.137 | 0.820 | 0.150 | 0.262 | 0.291 | 0.389 | 0.402 | 0.457 | 0.452 | 0.504 | **0.114** | **0.227** |
| PEMS07 | 12 | **0.062** | **0.162** | 0.075 | 0.183 | 0.118 | 0.235 | 0.067 | 0.165 | 0.095 | 0.207 | 0.094 | 0.200 | 0.115 | 0.242 | 0.068 | 0.171 |
| | 24 | **0.086** | 0.192 | 0.113 | 0.225 | 0.242 | 0.341 | 0.088 | **0.190** | 0.150 | 0.262 | 0.139 | 0.247 | 0.210 | 0.329 | 0.119 | 0.225 |
| | 48 | 0.128 | 0.234 | 0.157 | 0.254 | 0.562 | 0.541 | **0.110** | **0.215** | 0.253 | 0.340 | 0.311 | 0.369 | 0.398 | 0.458 | 0.149 | 0.237 |
| | 96 | 0.176 | 0.268 | 0.207 | 0.291 | 1.096 | 0.795 | **0.139** | 0.245 | 0.346 | 0.404 | 0.396 | 0.442 | 0.594 | 0.553 | 0.141 | **0.234** |
| PEMS08 | 12 | 0.082 | 0.185 | 0.091 | 0.201 | 0.133 | 0.247 | **0.079** | **0.182** | 0.168 | 0.232 | 0.165 | 0.214 | 0.154 | 0.276 | 0.087 | 0.184 |
| | 24 | 0.117 | 0.226 | 0.140 | 0.251 | 0.249 | 0.343 | **0.115** | **0.219** | 0.224 | 0.281 | 0.215 | 0.260 | 0.248 | 0.353 | 0.122 | 0.221 |
| | 48 | **0.169** | 0.268 | 0.200 | 0.291 | 0.569 | 0.544 | 0.186 | **0.235** | 0.321 | 0.354 | 0.315 | 0.355 | 0.440 | 0.470 | 0.189 | 0.270 |
| | 96 | 0.233 | 0.306 | 0.272 | 0.328 | 1.166 | 0.814 | **0.221** | **0.267** | 0.408 | 0.417 | 0.377 | 0.397 | 0.674 | 0.565 | 0.236 | 0.300 |
| Avg. | | 0.125 | 0.229 | 0.149 | 0.252 | 0.514 | 0.482 | **0.119** | **0.218** | 0.217 | 0.306 | 0.220 | 0.304 | 0.320 | 0.394 | 0.121 | 0.222 |

CycleNet, formed by combining the RCF technique with a simple backbone, achieved state-of-the-art performance across multiple domains but fell short in the traffic domain. To further investigate the reasons behind this, we supplemented the complete performance of CycleNet on the PEMS dataset (the same four public subsets adopted in SCINet [34]) in Table 11. The results show that: (i) CycleNet still achieved top-tier prediction accuracy, and (ii) although CycleNet underperformed compared to iTransformer in this scenario, the gap in MSE on the Traffic dataset was reduced from approximately 10% to about 5%.

Regarding the first point, it is important to highlight the effectiveness of RCF. CycleNet's backbone is merely a single-layer Linear or a two-layer MLP, without any additional design or deep stacking, yet it still delivers excellent results. Specifically, when comparing CycleNet/Linear with RLinear and DLinear, it becomes evident that RCF is the major contributor to narrowing the gap between the simple Linear model and those state-of-the-art models.

Table 12: Statistical characteristics of datasets, including average number of extreme points per channel (Z-Score > 6), average maximum extreme value per channel, and cosine similarity between channels.

| | Traffic | Electricity | Solar-Energy | ETTh1 | PEMS03 | PEMS04 | PEMS07 | PEMS08 |
|---|---|---|---|---|---|---|---|---|
| Avg. Extreme Points | **23.8** | 1.4 | 0 | 0 | 0.9 | 0.1 | 3.5 | 4.8 |
| Avg. Max Extreme | **9.27** | 4.14 | 2.92 | 4.08 | 2.87 | 2.66 | 2.61 | 2.77 |
| Cosine Similarity | 0.56 | 0.46 | **0.92** | 0.21 | 0.84 | 0.77 | 0.80 | 0.78 |

For the second point, we further analyzed the statistical characteristics of the datasets to explore the underlying reasons in Table 12. Specifically, we examined the presence of extreme values in the channels and the cosine similarity between channels. It was found that the Traffic dataset contains very significant outliers, both in terms of quantity and magnitude. The presence of these outliers:

(i) **May affect the effectiveness of RCF.** The fundamental working principle of RCF is to learn the historical average cycles in the dataset. In such cases, the average cycles learned in RCF can be skewed by these significant outliers, such as the mean of a certain point in the cycle being exaggerated.

Consequently, during each prediction process, the original sequence subtracts a locally exaggerated average cycle, resulting in an inaccurate residual component and affecting the local point predictions within each cycle. The more inaccurate these local point predictions are, the larger the discrepancy between MSE and MAE, as MSE significantly amplifies the impact of a few large errors. This explains why in Table 4, combining iTransformer with RCF decreases MAE but increases MSE, indicating overall prediction accuracy improvement but anomalies in local point predictions.

(ii) **Highlight the necessity of stronger spatiotemporal relationship modeling.** Models like iTransformer and GNN, which accurately model inter-channel relationships, are more suitable for scenarios with extreme points and temporal lag characteristics. For example, when a sudden traffic surge occurs at a certain junction, these models, having correctly modeled the spatiotemporal relationships, can accurately predict possible traffic surges at other junctions. In contrast, the current CycleNet only considers single-channel relationship modeling, making it somewhat limited in this scenario.

These underlying reasons explain why CycleNet did not achieve the best performance on the Traffic dataset and showed a relative large performance gap. On the PEMS dataset, although it is also a traffic dataset, the presence of extreme points is significantly less severe compared to the Traffic dataset. Therefore, CycleNet's performance on the PEMS dataset improved compared to the Traffic dataset (the gap in MSE compared to the state-of-the-art reduced from approximately 10% to about 5%). This further validates the effectiveness of RCF but also indicates that in more complex traffic scenarios, reasonable spatiotemporal relationship modeling (or multivariate relationship modeling) is essential.

Additionally, while intuitively the solar scenarios might also involve significant spatiotemporal relationships, in practice, these relationships are much weaker compared to the traffic scenarios. Firstly, the weather conditions in the same region are often similar, leading to similar power generation curves. For instance, the Solar-Energy dataset's channels have a cosine similarity as high as 0.92 (shown in Table 12), which indirectly indicates weaker spatial characteristics. Secondly, extreme points are rare in the solar scenarios because photovoltaic systems have a maximum power threshold. Fewer extreme points mean that the impact of temporal lag characteristics is smaller. This explains why, compared to the Traffic dataset, the gains from the RCF technique are much more significant on the Solar-Energy dataset.

In summary, when dealing with traffic scenarios that may involve significant outliers and emphasize spatiotemporal relationship modeling, the current version of CycleNet may not be fully adequate. There are two direct and meaningful directions for improvement that could address this issue: (1) Enhancing the current RCF technique to be more robust to the presence of outliers; (2) Exploring a more reasonable multi-channel modeling technique within the RCF framework. We leave these challenges for future work and encourage the community to further research more robust and powerful periodic modeling techniques.

