# OpenReview forum: "CycleNet: Enhancing Time Series Forecasting through Modeling Periodic Patterns"
_NeurIPS.cc/2024/Conference — NeurIPS 2024 spotlight_

### Official Review · Reviewer_EqMa · 2024-06-23

**Soundness:** 4
**Presentation:** 3
**Contribution:** 4
**Rating:** 8
**Confidence:** 4

**Summary:**

This paper proposes a novel and effective technique to enhance long-term time series forecasting. The proposed technique, Residual Cycle Forecasting (RCF), directly models periodic cycles with learnable parameters, decomposing the learning of time series into periodic cycles and residual components. The residual can be learned with a simple architecture like Linear or MLP, and this technique can be integrated into existing forecasting models. Results show significant improvement achieved by using this technique.

**Strengths:**

* The proposed method is simple and effective. It not only improves accuracy but is also more parameter efficient and can be easily integrated into different backbones.
* The paper is well-written with good clarity. Both the problem and analysis are presented clearly.
* The experiments are conducted with high quality, providing extensive and comprehensive analysis of the proposed technique. The results are consistent with prior related work. Limitation of the proposed method is also clearly stated.

**Weaknesses:**

The notations are not strictly consistent. The notations for instance normalization (in Section 3.2 and Algorithm 2) are independent from the whole framework and not consistent with the previous problem definition.

**Questions:**

1. Since Figure 2 depicts the entire CycleNet architecture, instance normalization should also be included.

2. The description of alignment and repetition of $Q$ (Lines 152-157) can be further improved. The link to Algorithm 1 should be added, and a schematic figure (even in Appendix) would make it clearer to understand.

3. How extreme points affect RCF in the Traffic dataset should be further explained (Line 255). Extreme points can also affect other methods. It is unclear why this would specifically deteriorate RCF.

**Limitations:**

See Weaknesses and Questions

---

> ### Author Rebuttal · Authors · 2024-08-06
>
> **Thank you for your valuable comment!**
>
> > **W1:** The notations are not strictly consistent. The notations for instance normalization (in Section 3.2 and Algorithm 2) are independent from the whole framework and not consistent with the previous problem definition.
>
> Thanks for pointing this out. **We will correct it to make the notations in the paper more consistent.** In the original version of our paper, we used simpler notations (e.g., $x_{in}$, $y_{out}$) in Section 3.2 and Algorithm 2 to help readers better understand the model's input-output workflow. However, this indeed led to inconsistency in the notations. *Therefore, we will properly revise these in the revised paper.*
>
> > **Q1:** Since Figure 2 depicts the entire CycleNet architecture, instance normalization should also be included.
>
> Thanks, and **we will include instance normalization in Figure 2** to present the complete workflow of CycleNet.
>
> > **Q2:** The description of alignment and repetition of Q (Lines 152-157) can be further improved. The link to Algorithm 1 should be added, and a schematic figure (even in Appendix) would make it clearer to understand.
>
> Thanks for your suggestion. In our submission, due to space limitations in the main text, we adopted a concise writing style. **We will further elaborate on the alignment and repetition of $Q$ in the revised paper**, including more textual explanations in the main text, adding the link to Algorithm 1, and most importantly, supplementing with an intuitive schematic figure to help readers better understand this process (as shown in ***Figure 1 of the attached pdf***).
>
> > **Q3:** How extreme points affect RCF in the Traffic dataset should be further explained (Line 255). Extreme points can also affect other methods. It is unclear why this would specifically deteriorate RCF.
>
> The reason behind this is indeed complex. Firstly, the basic fact is that **there are indeed some outliers with extremely high values in the Traffic dataset.** For instance, the 12630 value of the 857th channel in the Traffic dataset, after global standard normalization, still exceeds 25, while the normal range of values for surrounding points is [-2, 2]. Secondly, **the fundamental working principle of RCF is to learn the historical average cycles in the dataset.**
>
> In such cases, the average cycles learned in RCF can be affected by these significant outliers, *such as the mean of a certain point in the cycle being exaggerated.* Consequently, in each prediction process, the original sequence subtracts a locally exaggerated average cycle, resulting in an inaccurate residual component, thereby affecting the local point predictions within each cycle. The more inaccurate these local point predictions are, the larger the discrepancy between MSE and MAE, as MSE significantly amplifies the impact of a few large errors. **This explains why in Table 4, combining iTransformer with RCF decreases MAE but increases MSE, indicating overall prediction accuracy improvement but anomalies in local point predictions.**
>
> Therefore, models like iTransformer and GNN, which accurately model inter-channel relationships, are more suitable for scenarios with extreme points and temporal lag characteristics. For example, when a sudden traffic surge occurs at a certain junction, these models, having correctly modeled the spatiotemporal relationships, can accurately predict possible traffic surges at other junctions. In contrast, the current CycleNet only considers single-channel relationship modeling, thus being somewhat limited in this scenario.
>
> *However, **the core of CycleNet still lies in exploring a more effective periodic modeling approach and proposing a model that balances performance and efficiency.*** We believe that further addressing the issue of RCF being affected by extreme points and investigating how to incorporate channel relationship modeling within CycleNet in future work would be highly promising. *We will include these discussions in the revised paper.*
>
>  **Thank you again for your careful review, and we hope our responses address your concerns.**

---

> > ### Comment · Reviewer_EqMa · 2024-08-08
> >
> > Thank you for your detailed response. I have carefully reviewed your rebuttal and the attached PDF. All my questions have been clearly addressed. My rating has been updated.

---

> > > ### Author Response · Authors · 2024-08-08
> > >
> > > Thank you very much! We will incorporate the revisions mentioned in the rebuttal into the final paper. Once again, thank you for your careful review and for increasing our score!

---

### Official Review · Reviewer_zzRb · 2024-07-12

**Soundness:** 3
**Presentation:** 3
**Contribution:** 3
**Rating:** 6
**Confidence:** 3

**Summary:**

The paper introduces CycleNet, a novel time series forecasting method that enhances long-term prediction accuracy by explicitly modeling the inherent periodic patterns present in time series data. The core contribution of paper is introducing the Residual Cycle Forecasting (RCF) technique, which leverages learnable recurrent cycles to represent these periodic patterns and predicts the residuals, significantly improving upon the performance of existing models with reduced computational complexity. CycleNet demonstrates state-of-the-art results across various domains, such as electricity, weather, and energy forecasting, while offering over 90% reduction in parameter quantity, highlighting its efficiency and effectiveness in capturing long-term dependencies for accurate forecasting.

**Strengths:**

1. The author tries to model the period information explicitly in the time series prediction task, and the motivation is intuitive and reasonable.

2. The method designed by the author is reasonable and closely related to motivation. Combined with the experimental results, the author gives a simple but effective method.

3. The author exposes the code and gives a detailed description, which increases the reproducibility of the model.

4. The limitations of this method are clearly discussed and the possible problems are pointed out.

**Weaknesses:**

1. In the introduction, the author's statement establishes a close relationship between long-term prediction and periodic information. In the absence of some experimental support, this is not rigorous. Periodic information may be useful for long-term forecasting in certain situations, but it is not appropriate for all tasks, nor is it the only important information that these tasks require. The author slightly obfuscates these to highlight the motivation of this article.

2. The authors select data sets with different periodicity to illustrate the validity of the model. However, the authors only demonstrate the validity, and experiments can be added to show how the proposed method performs differently on periodically different data sets and discuss the underlying rules. At the same time, it is also worth showing how CycleNet performs on data sets where there is no obvious periodicity.

3. Combined with the results in Table 2 and Table 4, the results predicted with Linear alone are even better than some well-designed methods, whether this is due to the particularity of the data set or some other reason.

**Questions:**

Please refer to the weakness

**Limitations:**

Please refer to the weakness

---

> ### Author Rebuttal · Authors · 2024-08-06
>
> **Thank you for your kind and careful review!**
>
> > **W1:** In the introduction, the author's statement establishes a close relationship between long-term prediction and periodic information. Etc.
>
> **In fact, periodic information is indeed one of the most important factors for achieving long-term forecasting.** Recent works have demonstrated that by effectively utilizing periodic information, it is possible to achieve near state-of-the-art prediction accuracy with fewer than 1,000 parameters [1]. This strongly underscores the importance of periodic information in long-term forecasting tasks. Additionally, the popular work DLinear has shown that a single-layer linear model can outperform many well-designed models in long-term forecasting tasks [2]. This is because a simple linear layer can robustly extract periodic information from the historical data of a single channel (evident from the clear periodic patterns in the weight distribution of these linear models) [3].
>
> *Without periodicity, long-term forecasting becomes very challenging.* For instance, DLinear shows that on financial datasets, even the most advanced deep learning models cannot outperform simply copying the most recent data point [2].
>
> Thus, existing evidence highlights that the presence of periodicity in data is crucial for accurate long-term predictions. **Our paper builds on this premise, exploring a simple yet effective method of leveraging periodicity through RCF.** We acknowledge that periodic information is not the sole factor for accurate time series predictions. Other factors like short-term trends, multivariate dependencies, and seasonality are also critical, especially for short-term forecasting tasks. However, the core of this paper focuses on improving long-term forecasting performance by better utilizing periodicity in data. Therefore, this motivation and contribution here may be not in conflict with scenarios where other factors are more influential.
>
> > **W2:** The authors select data sets with different periodicity to illustrate the validity of the model. However, the authors only demonstrate the validity, and experiments can be added to show how the proposed method performs differently on periodically different data sets and discuss the underlying rules. At the same time, it is also worth showing how CycleNet performs on data sets where there is no obvious periodicity.
>
> *Sorry if we misunderstood your first point here.* We have already shown the performance of our model on datasets with different cycle lengths in Table 5 and Figure 3 of our submission, **indicating that when the model's hyperparameter $W$ is set correctly, the RCF technique significantly improves prediction accuracy, and the trainable recurrent cycle $Q$ can accurately learn the corresponding cycle patterns.**
>
> We assume that your question is about how the proposed method performs differently with different cycle lengths and we have supplemented *Figure 2 (m-p) in the attached PDF* to visualize the learned recurrent cycle $Q$ when setting different cycle lengths $W$. We found that when $W$ is correctly set to 144 (the weekly cycle length of the Electricity dataset), $Q$ learns the complete cycle pattern, including weekly and daily cycles. When $W$ is set to 24 (the daily cycle length), $Q$ captures only the daily cycle. With $W$ set to 96 (four times the daily cycle), $Q$ learns four repeated daily cycles. However, with $W$ set to 23 (with no matching semantic cycle length), $Q$ learns nothing, resulting in a flat line. For datasets without obvious periodicity (e.g., Exchange-Rate dataset), the behavior of $Q$ is similar, learning a flat line. *We will further include these results in the revised paper.*
>
> > **W3:** Combined with the results in Table 2 and Table 4, the results predicted with Linear alone are even better than some well-designed methods, whether this is due to the particularity of the data set or some other reason.
>
> The results in Table 2 and Table 4 cannot be directly compared because Table 2 reports results averaged across all prediction horizons $H ∈ {96, 192, 336, 720}$, whereas Table 4 presents results for specific horizons individually.
>
> However, if we average the results from Table 4, *the Linear model does indeed outperform some well-designed models*, such as Autoformer. This phenomenon can be traced back to the DLinear paper [2], **which demonstrated that a purely linear layer can outperform many well-designed Transformer models at the time**. DLinear's success is attributed to its channel-independent approach for multivariate forecasting, where each channel is modeled with a shared linear layer. This approach allows the model to focus on extracting historical information from individual channels, leading to more robust periodic information for long-term forecasting (as evidenced by the striped patterns in the learned weights of its linear layers). Previous models like Autoformer mixed information across multiple channels, making it difficult to extract robust periodic information. Since DLinear, many models have adopted the channel-independent approach for long-term forecasting, including PatchTST, FITS, SparseTSF, and our CycleNet.
>
> **Therefore, it is reasonable that a simple linear model can outperform some well-designed models in this context.**
>
> [1] Lin, Shengsheng, et al. "SparseTSF: Modeling Long-term Time Series Forecasting with *1k* Parameters." In International Conference on Machine Learning, 2024.
>
> [2] Zeng, Ailing, et al. "Are transformers effective for time series forecasting?." Proceedings of the AAAI conference on artificial intelligence. Vol. 37. No. 9. 2023.
>
> [3] Toner, William, and Luke Darlow. "An Analysis of Linear Time Series Forecasting Models." In International Conference on Machine Learning, 2024.
>
> **Thank you again for your kind review, and we hope our response can address your concerns.**

---

> > ### Author Response · Authors · 2024-08-12
> >
> > **Dear Reviewer zzRb,**
> >
> > Sorry to bother you. **We are eager to know whether our response has addressed your concerns as the discussion phase is nearing its end.** *If not, or if you have any additional questions, we would be more than happy to further address them.* Thank you for your time.

---

> > > ### Comment · Reviewer_zzRb · 2024-08-12
> > >
> > > Thanks for the response. I will keep my original score.

---

> > > > ### Author Response · Authors · 2024-08-12
> > > >
> > > > Dear reviewer, thank you for your timely response. Once again, we sincerely thank you for taking the time to review our paper!

---

### Official Review · Reviewer_24Zo · 2024-07-12

**Soundness:** 3
**Presentation:** 3
**Contribution:** 2
**Rating:** 8
**Confidence:** 5

**Summary:**

This paper presents a novel technique for improving the accuracy of multivariate long-term time series forecasting. The technique, called Residual Cycle Forecasting (RCF), involves learning the cyclical patterns of time series through recurrent cycles, which can be used as a pre-processing step for any forecasting model. The authors also propose CycleNet, a linear-based model that uses RCF to enhance its predictions. CycleNet first uses RevIn to account for distribution shift, then subtracts the learned RCF from the input data. The backbone of the model predicts the future residual, adds the learned RCF, and reverses RevIN from the outputs to obtain the final prediction.

The proposed method is evaluated on eight multivariate time series datasets and compared against several baselines. The results show competitive performance and resource consumption.

**Strengths:**

* Clear and concise writing style
 * Novel approach to time series decomposition
 * Comparison of performance and resource consumption with baseline methods
 * Ablation study and parameter impact analysis (e.g., $W$)
 * Easy-to-understand model design and components
 * Informative figures to illustrate model and results
 * Thorough discussion of results, including strengths and limitations
 * Code and data provided for reproducibility and transparency.

**Weaknesses:**

* Lack of clarity to specify which results are from the authors (reproduced or produced) and which ones are collected from previous papers (if so, which ones)
 * Incomplete comparison with existing time series decomposition baselines, such as LD, TDFNet, or SparseTSF (even though this one was in the related works)
 * Model was not compared to RLinear (also relying on RevIN)

**Questions:**

## Baselines

The paper fails to properly compare the proposed CycleNet model with appropriate baselines.

Since CycleNet is based on RevIN, it would be more complete to include other RevIN-based baselines such as RLinear [1] for linear-based models and for instance RevInformer (from the RevIN paper) for transformer-based solutions. Although PatchTST uses RevIN, including these additional baselines would avoid any doubt and provide a more comprehensive comparison.

Additionally, the paper misses important baselines such as LD [2], which also uses RevIn and also argues for using learnable decomposition rather than moving average. According to the published results, CycleNet appears to be behind LD. But it would be interesting to compare LD against backbone + RCF.
Potential references to [3] is also missing.

[1] https://arxiv.org/pdf/2305.10721

[2] https://arxiv.org/abs/2402.12694

[3] https://arxiv.org/pdf/2308.13386

Finally, despite being describes as main baselines, why SparseTSF was not included in this paper? Especially, as SparseTSF seems to produce better results than proposal especially for ETT and not for Electricity and Traffic. What would be the reason for such differences?

Including such comparisons is required to correctly position CycleNet with all the other baselines and further discuss why CycleNet offers better performance with some datasets and not others. Such discussion is important to further investigate the benefit of time series decomposition for forecasting tasks.

## Visualization
The visualizations of the learned cycles are informative, but it is unclear whether these cycles change depending on the prediction horizon or lookback. If they do, it would be interesting to see the differences and whether they can explain the variation in performance for different prediction horizons.

Since the cycles are learned along with the backbone model, there could be differences in the learned cycles for different experiments. It would be helpful to plot the cycles for different prediction lengths, such as 96 and 720, and for different lookback lengths, such as 48, 96, and 336.

In addition, the cycle may change depending on the backbone. It would be interesting to show the difference if any when RCF is used with iTransformer, DLinear, etc. This would allow readers to better understand whether there are any differences in the learned cycles and whether they are due to the parameters, backbone model or other factors.

Figure 3(f) is surprising, and it is unclear whether it represents a household or an industrial building with specific working hours. In my opinion, it looks like there are only 3 main weekly patterns (the large pattern on the left seems to have ~50 time steps = 2 days) and weekends (one pattern seems to have ~24 time steps = 1 day). Therefore, it could be a factory or industrial building with specific working hours (6 days a week).

It would be interesting to discuss how RCF impacts previous backbones, such as DLinear. Specifically, it would be helpful to know how much DLinear's performance changes with the addition of RCF and plot it similarly to Figure 4.

Finally, it is important to provide Figure 4 and Table 4 for other datasets to give readers a global view of the impact of RCF/CycleNet depending on the dataset and its nature. This would help avoid generalizations that could be incorrect based only on results depicted in the current version, especially for datasets such as Traffic and Solar Energy where the impact might be drastically different.

## Discussion
“Traffic dataset exhibits spatiotemporal characteristics and temporal lag characteristics, where the traffic flow at a certain detection point significantly affects the future values of neighboring detection points.” However, the authors do not explain why this is not an issue for the Solar-Energy dataset, where weather conditions at one location may slowly impact neighboring locations in the future. Can authors comment on that?

Regarding the electricity consumption dataset, the authors state that "a user’s electricity consumption thirty days ahead does not directly correlate with their consumption patterns in the past few days." However, it is possible that long-term habits may still influence consumption patterns, for instance with a monthly routine.

The authors suggest that "the cycle length W depends on the a priori characteristics of the dataset and should be set to the maximum stable cycle within the dataset." However, they do not address how to account for yearly cycles, such as those found in weather datasets that repeat each year with the seasons or datasets influenced by human behavior, such as electricity consumption, which may increase in winter and summer due to heating and cooling needs. It would be helpful to clarify whether RCF is only suitable for mid-range stable cycles or if there is a way to account for longer cycles. This might need to be included in the limitations section.

## Additional points
The paper lacks a computation cost study to determine the overhead imposed by RCF on the existing backbone in terms of memory and consumption. It is important to evaluate whether the gain in accuracy justifies the increase in complexity and associated cost.

## Proof-read
 * “a single-layer Linear or a dual-layer MLP” maybe find another way to express it to avoid repeating too much (cf., abstract, introduction, etc.)
 * “surpassing complexly designed deep models”->  redundant, ”complex design” or “deep models”
 * “leveraging this discovery to enhance” should change this as the fact that time series have cycle is not new leveraging this knowledge is the innovation

**Limitations:**

Authors have discussed some limitations of their proposal and especially when each channel have different period cycle.

---

> ### Author Rebuttal · Authors · 2024-08-06
>
> **Thank you for your detailed and thoughtful review!**
>
> >  **W1:** Lack of clarity of the source of results.
>
> We will clarify this in the main text of the revised paper. Previously, we clarified this in Appendix A.4.
>
> > **W2 & W3 & Q1: Baseline:** Add more appropriate baselines to correctly position the RCF technique.
>
> Thank you for your reminder.
>
> (i) **Leddam (LD) and SparseTSF** are both accepted papers at ICML 2024. We timely noted the latter because it emphasizes the importance of periodicity, similar to our work. LD upgrades the Moving Average kernel (MOV) technique in STD to a weights-learnable module, while SparseTSF uses sparse techniques to decompose sequences. We will compare these two techniques with our proposed RCF technique.
>
> (ii) **TFDNet** extracts features in the Time-Frequency domain after using STD techniques to decompose sequences. Since it fundamentally uses regular MOV-based STD techniques, it may not be necessary to compare it directly here.
>
> (iii) **RLinear**: We will add a comprehensive comparison with it in the main text. In fact, the Linear and MLP in Table 4 are combined with RevIN, so they represent RLinear and RMLP, and CycleNet significantly outperforms them. Additionally, we thoroughly evaluated the impact of RevIN on CycleNet in Appendix Table 8.
>
> **In summary, we compared CycleNet/Linear (RCF+Linear), LDLinear (LD+Linear), DLinear (MOV+Linear), SparseTSF (Sparse+Linear), and Linear** in *Table 1 of the attached pdf*. To ensure fairness, we did not use RevIN since DLinear originally did not use it.
>
> It can be observed that:
>
> (i) **CycleNet significantly outperforms other methods**, proving the superiority of RCF as a new STD technique.
>
> (ii) As a sparse prediction method, SparseTSF may rely more on longer lookback length and RevIN, thus performing poorly here.
>
> (iii) Most surprisingly, **the performance of LDLinear and DLinear is almost identical** in our setup. This result differs from the results reported in the LD paper (Table 4). Other researchers have also noted this phenomenon on the LD project's GitHub (Issue #1), but have not yet received a response from the authors. *To some extent, LD is essentially a weight-trainable MOV, so it is hard to achieve high performance gains compared to the original MOV.* We are not sure if there is any mistake here, so we will further confirm with the authors after the double-blind review phase.
>
> > **Q2: Visualization:** Add more plots under different configurations.
>
> **We have added these visualizations** in *Figure 2 of the attached pdf*. The basic configuration is Electricity-Lookback=96-Horizon=96-Backbone=Linear-W=168. It can be observed that:
>
> **(i) Horizon:** *The learned patterns remain almost unchanged as the horizon changes.* This indicates that the horizon length does not affect the learned pattern results.
>
> **(ii) Lookback:** The overall pattern remains unchanged as lookback changes. However, upon closer observation, *it can be seen that the learned pattern becomes smoother with increased lookback.* This is because a longer lookback provides the backbone with richer periodic information, thereby reducing the importance of the learned pattern component.
>
> **(iii) Backbone:** *The patterns change somewhat with different backbones.* When DLinear is the backbone, the learned patterns are smoother, as DLinear's decomposition technique itself extracts certain periodic features. When iTransformer is the backbone, the learned patterns differ more, as it additionally models multichannel relationships, so the learned periodic patterns may consider multichannel feature interactions. PatchTST's performance is more similar to Linear, as it is also a regular single-channel modeling method, only with stronger nonlinear learning capabilities compared to the Linear model.
>
> **(iv) Cycle length $W$:** When $W$ is 144 (the weekly cycle length for the Electricity dataset), the recurrent cycle $Q$ learns the complete periodic pattern, including weekly and daily cycles. When $W$ is set to 24 (the daily cycle length), the recurrent cycle $Q$ only learns the daily cycle pattern. When $W$ is set to 96 (four times the daily cycle length), the recurrent cycle $Q$ learns four repeated daily cycle models. However, when $W$ is set to 23 (without matching semantic meaning), the recurrent cycle $Q$ learns nothing, i.e., a straight line.
>
> > **Q3: Discussion and other points.**
>
> *Very sorry that the response here cannot cover every point you have mentioned **due to the text length limitation***. We will carefully incorporate your suggestions into the revision (e.g., providing more results of other datasets on Table 4 and Figure 4), and **we can further discuss any uncovered points during the discussion phase.**
>
> > **Q4: Additional points:** The computation cost of RCF on other Backbones.
>
> **In fact, the cost of RCF is fixed and independent of the Backbone.** The primary additional fixed overhead is the parameter quantity of $D×W$, and the CPU time required for alignments and repetitions of the recurrent cycles (~10 seconds per epoch in our experimental environment). **Overall, this additional overhead is very small compared to the other deep backbones.** We will supplement this quantitative analysis in Table 3 (i.e., the additional overhead introduced by RCF). By combining these overheads and the ablation results of other backbones in Table 4, it can be inferred whether the gain in accuracy justifies the increase in complexity and associated cost.
>
> > **Q5: Proof-read.**
>
> Thank you, and we will revise these points in the revised paper.
>
> **Again, thank you for your thorough review, and we hope our responses address your concerns.**

---

> > ### Author Response · Authors · 2024-08-08
> > **Additional Rebuttal (Optional Reading) # Part 1**
> >
> > **Dear Reviewer 24Zo,**
> >
> > Thanks again for you careful review. **It’s the discussion phase now, so we would like to continue to address the uncovered points from the rebuttal phase** (*due to the text length limits in the rebuttal*). *If you find these additional comments any inappropriate (e.g., out of rebuttal limits), please ignore these comments.*
> >
> > > Finally, despite being describes as main baselines, why SparseTSF was not included in this paper? Especially, as SparseTSF seems to produce better results than proposal especially for ETT and not for Electricity and Traffic. What would be the reason for such differences?
> >
> > As mentioned in the rebuttal, we noticed SparseTSF, which was recently accepted at ICML 2024, because it emphasizes the importance of periodicity in long-term forecasting tasks, similar to our work. Thus, we included it in the related work but did not perform a complete comparison due to the submission deadline constraints. We will supplement a complete comparison in the revised paper.
> >
> > Additionally, SparseTSF performs better on the ETT dataset rather than the Electricity and Traffic datasets for two reasons. First, **the ETT dataset is smaller and noisier**. In such cases, the sparse techniques proposed in SparseTSF help the model focus more directly on corresponding elements in historical cycles, reducing interference from less relevant elements and improving overall performance. Second, the ETT dataset's dimensionality (number of channels) is significantly lower than that of the Electricity and Traffic datasets, with the former having only 7 channels compared to the latter's 321 and 862 channels. Note that SparseTSF fundamentally employs a channel-independent modeling approach (parameter-shared) and linear-based methods, which limit its performance on high-dimensional datasets. This is because such scenarios require non-linear capabilities to remember different patterns of multiple channels, or a separate linear layer for each channel (non-parameter-shared) to model each channel's patterns separately [1]. *In contrast, CycleNet’s RCF technique is a fully channel-independent modeling scheme (non-parameter-shared)*, meaning it models each channel's periodic pattern separately. In this case, even with Linear as the backbone, it performs better on high-dimensional datasets because it enhances the model's ability to capture different patterns of multiple channels. Therefore, as shown in our comparison results in *Table 1 of the attached PDF*, **our RCF technique overall still outperforms the Sparse technique** (even on the ETT dataset). *We will include full experiments on more datasets in the revised paper.*
> >
> > [1] Li, Zhe, et al. "Revisiting long-term time series forecasting: An investigation on linear mapping." *arXiv preprint arXiv:2305.10721* (2023).
> >
> > > Figure 3(f) is surprising, and it is unclear whether it represents a household or an industrial building with specific working hours. In my opinion, it looks like there are only 3 main weekly patterns (the large pattern on the left seems to have ~50 time steps = 2 days) and weekends (one pattern seems to have ~24 time steps = 1 day). Therefore, it could be a factory or industrial building with specific working hours (6 days a week).
> >
> > Yes, you are correct. This might represent the typical working hours of a user, for example, working on Monday, Wednesday, and Friday, with the other days off. Indeed, this interesting phenomenon further reveals *the potential value of our proposed RCF technique*, as it may be **a superior way to help data engineers analyze patterns in time series data**.
> >
> > > “Traffic dataset exhibits spatiotemporal characteristics and temporal lag characteristics, where the traffic flow at a certain detection point significantly affects the future values of neighboring detection points.” However, the authors do not explain why this is not an issue for the Solar-Energy dataset, where weather conditions at one location may slowly impact neighboring locations in the future. Can authors comment on that?
> >
> > The Traffic and Solar-Energy datasets are indeed quite different.

---

> > > ### Author Response · Authors · 2024-08-08
> > > **Additional Rebuttal (Optional Reading) # Part 2**
> > >
> > > **First, the spatial characteristics of the solar scenario are weaker than those of the traffic scenario.** In the traffic scenario, traffic flow dynamically changes at each location, with complex internal spatial dependencies. In the solar power generation scenario, as in the Solar-Energy dataset, which records the solar power production of 137 PV plants in Alabama State, the spatial dependencies are much weaker because the weather conditions within a region are usually similar. Even if there are spatial relationships, such as differences in power generation due to longitude differences, they are minor and easily learned. The following table shows the average cosine similarity between different channels in the training set of each dataset (closer to 1 indicates more similar channels):
> > >
> > > | Dataset               | Traffic | Electricity | Solar-Energy | ETTh1 |
> > > | --------------------- | ------- | ----------- | ------------ | ----- |
> > > | **Cosine Similarity** | 0.578   | 0.471       | **0.913**    | 0.261 |
> > >
> > > *It can be seen that the power generation curves of different channels in the Solar-Energy dataset are very similar, which indirectly indicates weaker spatial characteristics.*
> > >
> > > **Second, the solar scenario has fewer extreme points compared to the traffic scenario, so the impact of temporal lag characteristics is smaller.** In the traffic scenario, unexpected situations may cause a sudden increase in flow (i.e., extreme points), and a traffic surge at one intersection can affect the flow at other intersections over time. In this case, adequately modeling inter-channel relationships (i.e., temporal lag) can accurately predict traffic flows at other intersections after an extreme point occurs. In contrast, extreme points are rare in the solar scenario because the power generation of photovoltaic systems has a maximum power threshold. The following table shows the average number of extreme points per channel using Z-Score > 6:
> > >
> > > | Dataset            | Traffic | Electricity | Solar-Energy | ETTh1 |
> > > | ------------------ | ------- | ----------- | ------------ | ----- |
> > > | **Extreme Points** | 63.9    | 0.5         | 0            | 0     |
> > >
> > > *It can be seen that the number of extreme points in the Traffic dataset is significantly higher than in other datasets.*
> > >
> > > > Regarding the electricity consumption dataset, the authors state that "a user’s electricity consumption thirty days ahead does not directly correlate with their consumption patterns in the past few days." However, it is possible that long-term habits may still influence consumption patterns, for instance with a monthly routine.
> > >
> > > Thank you for pointing this out. What we intended to convey is more along the lines of "a user’s electricity consumption thirty days ahead ***not only*** correlates with their consumption patterns in the past few days." That is, to achieve accurate long-term predictions, the long-term patterns play a crucial role, not just the short-term fluctuations. **We will revise the original statement.**
> > >
> > > > The authors suggest that "the cycle length W depends on the a priori characteristics of the dataset and should be set to the maximum stable cycle within the dataset." However, they do not address how to account for yearly cycles, such as those found in weather datasets that repeat each year with the seasons or datasets influenced by human behavior, such as electricity consumption, which may increase in winter and summer due to heating and cooling needs. It would be helpful to clarify whether RCF is only suitable for mid-range stable cycles or if there is a way to account for longer cycles. This might need to be included in the limitations section.
> > >
> > > Thank you for your suggestion. **We will include this in the limitations section.** Considering longer dependencies (such as yearly cycles) is indeed a more challenging task. Although theoretically, CycleNet’s $W$ can be set to a yearly cycle length to model annual cycles, the biggest difficulty lies in collecting sufficiently long historical data to train a complete yearly cycle (possibly requiring decades of data). For example, the Electricity training set covers only 2 years, and the Weather dataset includes less than a year. In this case, other existing models might also struggle to effectively model such long cycles. Therefore, CycleNet is more suitable for mid-range stable cycle modeling, and we believe future research needs to develop more advanced techniques to address this issue specifically.
> > >
> > > **Thank you again for your nice review.**

---

> > > > ### Author Response · Authors · 2024-08-12
> > > >
> > > > **Dear Reviewer 24Zo,**
> > > >
> > > > Sorry to bother you. **We are eager to know whether our response has addressed your concerns as the discussion phase is nearing its end.** *By the way, we would like to further include the remaining results of other datasets for Table 1 of the attached PDF,* which were not included initially due to space constraints. Thank you for your time.
> > > >
> > > > |             |      | RCF+Linear |           | LD+Linear |           | MOV+Linear |           | Sparse+Linear |           |  Linear   |           |
> > > > | :---------: | :--: | :--------: | :-------: | :-------: | :-------: | :--------: | :-------: | :-----------: | :-------: | :-------: | :-------: |
> > > > |             |      |    MSE     |    MAE    |    MSE    |    MAE    |    MSE     |    MAE    |      MSE      |    MAE    |    MSE    |    MAE    |
> > > > |    ETTh2    |  96  |   0.308    |   0.369   | **0.292** | **0.357** |   0.297    |   0.362   |     0.340     |   0.389   |   0.305   |   0.368   |
> > > > |             | 192  |   0.382    |   0.416   | **0.372** | **0.409** |   0.398    |   0.426   |     0.379     |   0.413   |   0.385   |   0.419   |
> > > > |             | 336  |   0.454    |   0.465   |   0.479   |   0.480   |   0.496    |   0.489   |   **0.404**   | **0.437** |   0.458   |   0.470   |
> > > > |             | 720  | **0.661**  | **0.575** |   0.675   |   0.582   |   0.694    |   0.592   |     0.720     |   0.600   |   0.691   |   0.592   |
> > > > |    ETTm1    |  96  | **0.298**  |   0.350   |   0.305   |   0.350   |   0.309    |   0.356   |     0.306     | **0.349** |   0.305   |   0.349   |
> > > > |             | 192  | **0.330**  |   0.370   |   0.335   | **0.366** |   0.346    |   0.380   |     0.339     |   0.370   |   0.338   |   0.369   |
> > > > |             | 336  | **0.359**  | **0.388** |   0.372   |   0.390   |   0.373    |   0.391   |     0.372     |   0.389   |   0.371   |   0.389   |
> > > > |             | 720  | **0.410**  | **0.421** |   0.445   |   0.443   |   0.439    |   0.435   |     0.430     |   0.426   |   0.433   |   0.428   |
> > > > |    ETTm2    |  96  | **0.164**  |   0.260   |   0.165   | **0.257** |   0.165    |   0.257   |     0.177     |   0.272   |   0.166   |   0.259   |
> > > > |             | 192  | **0.225**  | **0.304** |   0.240   |   0.318   |   0.232    |   0.310   |     0.246     |   0.325   |   0.228   |   0.305   |
> > > > |             | 336  | **0.271**  | **0.332** |   0.290   |   0.349   |   0.295    |   0.356   |     0.309     |   0.370   |   0.275   |   0.334   |
> > > > |             | 720  |   0.406    |   0.423   | **0.396** | **0.419** |   0.427    |   0.442   |     0.427     |   0.440   |   0.407   |   0.425   |
> > > > | Electricity |  96  | **0.131**  | **0.228** |   0.140   |   0.237   |   0.140    |   0.237   |     0.148     |   0.243   |   0.140   |   0.238   |
> > > > |             | 192  | **0.145**  | **0.242** |   0.154   |   0.250   |   0.154    |   0.250   |     0.159     |   0.254   |   0.154   |   0.251   |
> > > > |             | 336  | **0.160**  | **0.260** |   0.170   |   0.268   |   0.169    |   0.268   |     0.173     |   0.271   |   0.170   |   0.269   |
> > > > |             | 720  | **0.193**  | **0.292** |   0.204   |   0.300   |   0.204    |   0.301   |     0.207     |   0.303   |   0.204   |   0.301   |
> > > > |    Solar    |  96  | **0.192**  | **0.251** |   0.222   |   0.294   |   0.222    |   0.298   |     0.226     |   0.296   |   0.224   |   0.302   |
> > > > |             | 192  | **0.218**  | **0.258** |   0.249   |   0.315   |   0.250    |   0.312   |     0.252     |   0.312   |   0.250   |   0.310   |
> > > > |             | 336  | **0.231**  | **0.262** |   0.268   |   0.326   |   0.270    |   0.335   |     0.270     |   0.326   |   0.269   |   0.325   |
> > > > |             | 720  | **0.239**  | **0.265** |   0.271   |   0.327   |   0.272    |   0.327   |     0.271     |   0.327   |   0.270   |   0.333   |
> > > > |   Traffic   |  96  | **0.397**  | **0.275** |   0.411   |   0.285   |   0.411    |   0.284   |     0.414     |   0.281   |   0.411   |   0.283   |
> > > > |             | 192  | **0.412**  | **0.282** |   0.423   |   0.288   |   0.423    |   0.289   |     0.425     |   0.285   |   0.423   |   0.289   |
> > > > |             | 336  | **0.426**  | **0.290** |   0.436   |   0.296   |   0.436    |   0.296   |     0.436     |   0.293   |   0.437   |   0.297   |
> > > > |             | 720  | **0.456**  | **0.308** |   0.466   |   0.315   |   0.466    |   0.316   |     0.464     |   0.310   |   0.466   |   0.316   |
> > > > |   Weather   |  96  | **0.174**  |   0.240   |   0.174   | **0.235** |   0.177    |   0.241   |     0.177     |   0.238   |   0.176   |   0.238   |
> > > > |             | 192  |   0.218    |   0.279   |   0.227   |   0.292   |   0.222    |   0.283   |     0.224     |   0.287   | **0.217** | **0.276** |
> > > > |             | 336  |   0.262    |   0.314   |   0.270   |   0.326   | **0.261**  | **0.312** |     0.264     |   0.316   |   0.271   |   0.326   |
> > > > |             | 720  |   0.328    |   0.367   |   0.341   |   0.386   |   0.330    |   0.370   |   **0.326**   |   0.366   |   0.333   |   0.373   |

---

> > > > ### Comment · Reviewer_24Zo · 2024-08-13
> > > >
> > > > > Added visualizations
> > > >
> > > > Thank you, these are very interesting results. Again, make sure to include them as it might be very useful for other researchers tackling cyclical patterns or decomposition of time series data.
> > > >
> > > > > We will include full experiments on more datasets in the revised paper.
> > > >
> > > > Much appreciated.
> > > >
> > > > > It can be seen that the power generation curves of different channels in the Solar-Energy dataset are very similar, which indirectly indicates weaker spatial characteristics.
> > > >
> > > > > It can be seen that the number of extreme points in the Traffic dataset is significantly higher than in other datasets.
> > > >
> > > > These are very compelling explanations. Thank you for computing them and enlightening me as well as future readers.
> > > > The discussion regarding the limitations of existing solutions, as well as your proposal, is very interesting. Please ensure that you include them in your final revision.
> > > >
> > > > > We will include this in the limitations section.
> > > >
> > > > Thank you!

---

> > > > > ### Author Response · Authors · 2024-08-13
> > > > >
> > > > > Dear reviewer, thank you very much! We will do our best to incorporate the mentioned new results and analyses into the main text or the appendix. Once again, we are very grateful for your detailed and constructive review!

---

> > ### Comment · Reviewer_24Zo · 2024-08-13
> >
> > Thank you for your extensive effort during this rebuttal phase and for providing such detailed responses to my comments and those of other reviewers.
> > Overall, the authors have addressed all my points, and I still do not see any issues preventing the acceptance of this paper.
> > Even though it is a difficult task, please ensure that all the outputs provided during the rebuttal and discussion phase are added to the main paper (or at least a quick summary of them, with a reference to the appendix). These details are crucial for future readers to fully understand your work, its advantages compared to existing solutions, its limitations, and areas that require further investigation.
> >
> > In light of this, I have increased my score from 7 to 8.

---

> ### Comment · Reviewer_24Zo · 2024-08-13
>
> > Other researchers have also noted this phenomenon on the LD project's GitHub (Issue #1), but have not yet received a response from the authors.
>
> I was not aware of the existing issue with LD. It is good to include these results in your final revision and mention the existing reproducibility issue with LD. Such information will be helpful for future readers.

---

### Official Review · Reviewer_SdEc · 2024-07-13

**Soundness:** 3
**Presentation:** 2
**Contribution:** 3
**Rating:** 7
**Confidence:** 5

**Summary:**

This paper proposes a learnable Seasonal-Trend Decomposition method (CycleNet) to improve the prediction performance of current long-term multivariate time series forecasting models. Specifically, it firstly model the periodic patterns of sequences through globally shared recurrent cycles and then predicts the residual components of the modeled cycles. Extensive experiments are conducted to evaluate the proposed method.

**Strengths:**

1. The proposed method is a model-agnostic solution applicable to different kinds of models.
2. Although it is simple, it is able to achieve good performance improvement in many cases.
3. Extensive experiments are conduct to evaluate the proposal.

**Weaknesses:**

1. It seems that the proposal (CycleNet) does not work well for complex datasets, e.g., Traffic. It is better to show more results on the same kind of datasets like the PEMS datasets used in the iTransformer paper.
2. The Section 3 is not well written and lacks a lot of details, especially about the Learnable Recurrent Cycles. The authors may consider to reorganize Section 3 and Appendix A.1 to make Section 3 more clear.

**Questions:**

Please refer to the weaknesses.

**Limitations:**

Yes

---

> ### Author Rebuttal · Authors · 2024-08-06
>
> **Thank you for your valuable comment!**
>
> > **W1:** It seems that the proposal (CycleNet) does not work well for complex datasets, e.g., Traffic. It is better to show more results on the same kind of datasets like the PEMS datasets used in the iTransformer paper.
>
> The current CycleNet did not achieve SOTA on the Traffic dataset because **it is a simple single-channel modeling method that uses the proposed RCF technique to achieve a balance of performance and efficiency.** Its backbone for prediction is only a single Linear layer or a dual-layer MLP. For scenarios like Traffic, which exhibit spatiotemporal characteristics and temporal lag characteristics, more complex multi-channel relationship modeling is indeed required.
>
> *As we know, traffic scenarios may experience sudden surges in flow (i.e., extreme points), and the surge at one intersection can affect the flow at other intersections over time.* The table below shows the average number of extreme points per channel, counted with Z-Score > 6:
>
> | Dataset    | Traffic | Electricity | Solar | ETTh1 |
> | ---------- | ------- | ----------- | ----- | ----- |
> | Avg. Count | 63.9    | 0.5         | 0     | 0     |
>
> *We can see that the Traffic dataset has significantly more extreme points than other datasets.* In this case, models that fully model inter-channel relationships, like iTransformer, can accurately predict the flow at other intersections after an extreme point appears. In contrast, univariate models or over-parameterized multivariate models are less capable in such situations. This is why in Table 2, iTransformer significantly outperforms other models as it can accurately predict the flow at other intersections after an extreme event.
>
> **However, it is also notable that, aside from iTransformer, CycleNet still significantly outperforms other models**. Especially, this result is achieved with CycleNet's backbone being simple linear and MLP layers. Therefore, returning to CycleNet's core contribution, which is exploring a simpler yet effective use of periodicity and establishing a method that balances performance and efficiency, it is undoubtedly successful.
>
> Additionally, based on your suggestion, **we have added results of CycleNet on the PEMS datasets.** Here, it is essential to clarify that although PEMS and Traffic are both transportation datasets, PEMS has significantly fewer extreme points than Traffic:
>
> | Dataset    | Traffic | PEMS03 | PEMS04 | PEMS07 | PEMS08 |
> | ---------- | ------- | ------ | ------ | ------ | ------ |
> | Avg. Count | 63.9    | 0.9    | 0.1    | 3.5    | 4.8    |
>
>
> Thus, in this case, CycleNet is expected to perform better on PEMS than on Traffic. Below are the MSE comparison results of CycleNet and other models in the scenario where the lookback length is 96 and the prediction horizon is 12:
>
> | Dataset | CycleNet/MLP | CycleNet/Linear | iTransformer | PatchTST | Crossformer | TimesNet | DLinear | RLinear |
> | ------- | ------------ | --------------- | ------------ | -------- | ----------- | -------- | ------- | ------- |
> | PEMS03  | **0.066**    | 0.080           | 0.071        | 0.099    | 0.090       | 0.085    | 0.122   | 0.126   |
> | PEMS04  | **0.078**    | 0.089           | **0.078**    | 0.105    | 0.098       | 0.087    | 0.148   | 0.138   |
> | PEMS07  | **0.062**    | 0.075           | 0.067        | 0.095    | 0.094       | 0.082    | 0.115   | 0.118   |
> | PEMS08  | 0.082        | 0.091           | **0.079**    | 0.168    | 0.165       | 0.112    | 0.154   | 0.133   |
>
> **In this case, CycleNet/MLP performs on par with iTransformer, which models multi-channel relationships. Even the CycleNet/Linear, with a single linear layer as the backbone, outperforms other deep nonlinear models.** Therefore, these results further validate the effectiveness of the proposed RCF technique and demonstrate that the CycleNet model achieves a balance of performance and efficiency. *We will include these analyses and full experimental results in the revised paper.*
>
> > **W2:** The Section 3 is not well written and lacks a lot of details, especially about the Learnable Recurrent Cycles. The authors may consider reorganizing Section 3 and Appendix A.1 to make Section 3 clearer.
>
> Thank you for pointing out this issue. We will carefully optimize Section 3 in the revised paper to make it clearer for readers. Additionally, **we have supplemented it with a schematic figure** in *Figure 1 of the attached pdf*, which more intuitively describes the workflow of the recurrent cycle $Q$. *We will add this schematic figure to Section 3 of the revised paper.*
>
> **Thank you again for your valuable review, and we hope our response can address your concerns.**

---

> ### Comment · Reviewer_SdEc · 2024-08-09
>
> Thank you for the responses.  Could you give the full results on the PEMS dataset where the lookback length is 96 and the prediction horizon is {12, 24, 48, 96} (only the results of CycleNet, iTransformer and SCINet are OK).
>
> *Sorry, the prediction horizon should be  {12, 24, 48, 96} instead of {12, 24, 36, 48} according to Table 9 in iTransformer's paper.

---

> > ### Author Response · Authors · 2024-08-10
> >
> > **Dear Reviewer SdEc,**
> >
> > Thank you for your questions.
> >
> > We have provided the full results based on your requested settings. I apologize for the time it took to conduct these experiments. We will include the full results, including comparisons with other models, in the revised paper.
> >
> > As mentioned earlier, CycleNet is a simple, single-channel modeling method that uses only a Linear layer or a shallow MLP as the backbone. **Its purpose is to validate the proposed RCF technique, demonstrating that even with a simple backbone, combining RCF can achieve state-of-the-art prediction performance** (*except in the traffic scenario*) with very low computational overhead, balancing performance and efficiency.
> >
> > In scenarios like traffic, where spatiotemporal relationships need to be considered, independent channel modeling methods (including PatchTST, etc.) may struggle to fully capture the dynamics, necessitating additional multichannel relationship modeling techniques, such as those employed by iTransformer. Therefore, it is reasonable that the simple CycleNet has certain limitations in these spatiotemporal scenarios. *We had pointed out these limitations when analyzing the experimental results in the original submission and further elaborated on them in the limitations and future work sections.*
> >
> > In fact, iTransformer and SCINet are powerful models that achieve the best performance on the PEMS dataset. **The fact that CycleNet can nearly match their performance with just a simple backbone and independent channel modeling is noteworthy.** CycleNet’s backbone is merely a two-layer MLP, without any further design or deep stacking. (Due to length constraints, the results are attached in the next comments.)
> >
> > Moreover, *when the RCF technique is removed from CycleNet, its performance drops significantly*, demonstrating that RCF is a major contributor to narrowing the gap between the shallow MLP and these state-of-the-art models. *When the proposed RCF technique from CycleNet is integrated into iTransformer*, which already achieves state-of-the-art performance, *iTransformer’s predictive accuracy is further enhanced*.
> >
> > **Overall, for a basic two-layer MLP backbone, RCF brings about a 28% improvement in MSE and a 16% improvement in MAE.** **For iTransformer, which currently leads in the field, RCF brings an additional 4.9% improvement in MSE and 2.7% in MAE.** *This further validates the effectiveness of our RCF technique, which is our core contribution: a simple and novel method for better extracting periodicity in time series data.* In  addition to improving predictive accuracy, it can also serve as a novel decomposition method for helping us to further analyze the patterns present in time series data (as shown in Figure 3 in the paper).
> >
> > **I hope this addresses your concerns. Thank you again for your time and for reviewing our paper.**

---

> > > ### Author Response · Authors · 2024-08-10
> > >
> > > Due to length constraints, the results are attached here:
> > >
> > >
> > >
> > > |        |      | CycleNet/MLP  |           | iTransformer |           |  SCINet   |           | CycleNet W/o. RCF |       | iTransformer W/. RCF |           |
> > > | :----: | :--: | :-------: | :-------: | :----------: | :-------: | :-------: | :-------: | :---------------: | :---: | :------------------: | :-------: |
> > > |        |      |    MSE    |    MAE    |     MSE      |    MAE    |    MSE    |    MAE    |        MSE        |  MAE  |         MSE          |    MAE    |
> > > | PEMS03 |  12  |  *0.066*  |   0.172   |    0.071     |   0.174   |   0.066   |  *0.172*  |       0.077       | 0.186 |      **0.064**       | **0.170** |
> > > |        |  24  |   0.089   |   0.201   |    0.093     |   0.201   |  *0.085*  |  *0.198*  |       0.116       | 0.228 |      **0.084**       | **0.194** |
> > > |        |  48  |   0.136   |   0.247   |   *0.125*    |  *0.236*  |   0.127   |   0.238   |       0.181       | 0.289 |      **0.116**       | **0.228** |
> > > |        |  96  |   0.182   |   0.282   |   *0.164*    |  *0.275*  |   0.178   |   0.287   |       0.234       | 0.336 |      **0.163**       | **0.268** |
> > > | PEMS04 |  12  |   0.078   |   0.186   |    0.078     |   0.183   | **0.073** | **0.177** |       0.092       | 0.201 |       *0.075*        |  *0.182*  |
> > > |        |  24  |   0.099   |   0.212   |    0.095     |   0.205   | **0.084** | **0.193** |       0.133       | 0.248 |       *0.089*        |  *0.201*  |
> > > |        |  48  |   0.133   |   0.248   |    0.120     |   0.233   | **0.099** | **0.211** |       0.203       | 0.314 |       *0.110*        |  *0.225*  |
> > > |        |  96  |   0.167   |   0.281   |    0.150     |   0.262   | **0.114** | **0.227** |       0.257       | 0.357 |       *0.142*        |  *0.256*  |
> > > | PEMS07 |  12  | **0.062** | **0.162** |    0.067     |   0.165   |   0.068   |   0.171   |       0.073       | 0.177 |       *0.063*        | **0.162** |
> > > |        |  24  |  *0.086*  |   0.192   |    0.088     |  *0.190*  |   0.119   |   0.225   |       0.116       | 0.226 |      **0.078**       | **0.181** |
> > > |        |  48  |   0.128   |   0.234   |   *0.110*    |  *0.215*  |   0.149   |   0.237   |       0.201       | 0.301 |      **0.100**       | **0.202** |
> > > |        |  96  |   0.176   |   0.268   |   *0.139*    |   0.245   |   0.141   |  *0.234*  |       0.287       | 0.364 |      **0.126**       | **0.230** |
> > > | PEMS08 |  12  |   0.082   |   0.185   |   *0.079*    |  *0.182*  |   0.087   |   0.184   |       0.094       | 0.199 |      **0.076**       | **0.179** |
> > > |        |  24  |   0.117   |   0.226   |   *0.115*    |  *0.219*  |   0.122   |   0.221   |       0.151       | 0.255 |      **0.108**       | **0.213** |
> > > |        |  48  | **0.169** |   0.268   |   *0.186*    | **0.235** |   0.189   |   0.270   |       0.231       | 0.312 |        0.188         |  *0.238*  |
> > > |        |  96  |   0.233   |   0.306   |  **0.221**   |  *0.267*  |   0.236   |   0.300   |       0.332       | 0.380 |       *0.226*        | **0.265** |

---

> > > > ### Comment · Reviewer_SdEc · 2024-08-12
> > > >
> > > > Thank your for the results. I have updated my rating. Please include the full results for readers to have a more complete understanding of the proposal.

---

> > > > > ### Author Response · Authors · 2024-08-12
> > > > >
> > > > > Thank you very much! We will include the complete results in the final paper. Once again, thank you for taking the time to review our paper and for improving our rating!

---

### Author Rebuttal · Authors · 2024-08-06

**Dear AC and Reviewers,**

**Thank you very much for your time and effort in reviewing our submission.** The valuable comments provided are highly beneficial for improving the quality of our paper.

In this paper, **we propose the RCF technique**, which *utilizes learnable recurrent cycles to explicitly model the inherent periodic patterns within time series data*, followed by predicting the residual components of the modeled cycles. The RCF technique significantly enhances the performance of basic (or existing) models. **CycleNet** (which *combines RCF with basic Linear or MLP models*) **achieves consistent state-of-the-art performance** across multiple domains and offers **significant efficiency advantages**.

Overall, the four reviewers highly recognize the contributions of our submission, including comments such as "**Novel approach"**, "**Simple and effective**", "**Thorough discussion of results**", and "**Well-written with good clarity**". At the same time, the four reviewers have provided specific suggestions to improve the quality of our paper:

- **Reviewer SdEc** suggested supplementing experimental results on the PEMS dataset, and **we have supplemented these results** and demonstrated that the proposed method **remains very effective in this scenario**. In addition, Reviewer SdEc also suggested enhancing the description of the core technique proposed, and we will meticulously improve this part of the description in the final paper, and supplement an intuitive illustration to further explain the proposed technique (as shown in *Figure 1 of the attached pdf*).
- **Reviewer 24Zo** suggested adding comparisons with recent related work, and we have included this comparison in *Table 1 of the attached pdf*, **demonstrating the superiority of our methods**. Reviewer 24Zo also suggested showing more visualization results under different configurations, and **we have supplemented these results** in *Figure 2 of the attached pdf*. Additionally, Reviewer 24Zo raised more specific discussions and many detailed suggestions, including more discussion and some proof-reading. We will carefully revise the paper according to these suggestions.
- **Reviewer zzRb** suggested supplementing how the proposed method performs differently with different cycle lengths. **We have included this part of the experiment and discussion**. Additionally, Reviewer zzRb raised some issues about the introduction and results, and we have thoroughly analyzed these in the response.
- **Reviewer EqMa** provided suggestions for optimizing the writing, including notation consistency, figure completeness, and supplementary schematic figures. We will revise the paper and **have included the required schematic figure** in *Figure 1 of the attached pdf*. Additionally, Reviewer EqMa raised the issue of how extreme points affect the RCF technique, and we have **conducted an in-depth analysis**.

**Finally, thank you again for your valuable review.** We hope our response can further address your concerns, and we will carefully revise the paper according to the review.

Sincerely,

The Authors of Submission 9084

---

### Decision · Program_Chairs · 2024-09-25

**Decision:**

Accept (spotlight)

**Comment:**

This paper has been assessed by four knowledgeable reviewers who agreed on accepting it. Two of the reviewers gave it strong accept ratings, one marked it as straight accept and one as weak accept. The paper presents a novel yet simple approach to time series  decomposition that can be used for long term forecasting of time series. It is well written and well structured and avoids most of the common pitfalls in how the ideas are presented and supported with empirical evidence. The authors provided a thorough rebuttal and engaged the reviewers in focused discussions.